# ROBUST AND GENERALIZABLE VISUAL REPRESENTATION LEARNING VIA RANDOM CONVOLUTIONS

**Zhenlin Xu**[1], **Deyi Liu**[1], **Junlin Yang**[2], **Colin Raffel**[1], and **Marc Niethammer**[1]

[1] University of North Carolina at Chapel Hill
[2] Yale University
[1]{zhenlinx, mn, craffel}@cs.unc.edu, deyi@live.unc.edu
[2]junlin.yang@yale.edu

## ABSTRACT

While successful for various computer vision tasks, deep neural networks have shown to be vulnerable to texture style shifts and small perturbations to which humans are robust. In this work, we show that the robustness of neural networks can be greatly improved through the use of random convolutions as data augmentation. Random convolutions are approximately shape-preserving and may distort local textures. Intuitively, randomized convolutions create an infinite number of new domains with similar global shapes but random local texture. Therefore, we explore using outputs of multi-scale random convolutions as new images or mixing them with the original images during training. When applying a network trained with our approach to unseen domains, our method consistently improves the performance on domain generalization benchmarks and is scalable to ImageNet. In particular, in the challenging scenario of generalizing to the sketch domain in PACS and to ImageNet-Sketch, our method outperforms state-of-art methods by a large margin. More interestingly, our method can benefit downstream tasks by providing a more robust pretrained visual representation. [1]

## 1 INTRODUCTION

Generalizability and robustness to out-of-distribution samples have been major pain points when applying deep neural networks (DNNs) in real world applications (Volpi et al., 2018). Though DNNs are typically trained on datasets with millions of training samples, they still lack robustness to domain shift, small perturbations, and adversarial examples (Luo et al., 2019). Recent research has shown that neural networks tend to use superficial features rather than global shape information for prediction even when trained on large-scale datasets such as ImageNet (Geirhos et al., 2019). These superficial features can be local textures or even patterns imperceptible to humans but detectable to DNNs, as is the case for adversarial examples (Ilyas et al., 2019). In contrast, image semantics often depend more on object shapes rather than local textures. For image data, local texture differences are one of the main sources of domain shift, e.g., between synthetic virtual images and real data (Sun & Saenko, 2014). Our goal is therefore to learn visual representations that are invariant to local texture and that generalize to unseen domains. While texture and color may be treated as different concepts, we follow the convention in Geirhos et al. (2019) and include color when talking about texture.

We address the challenging setting of robust visual representation learning from *single domain data*. Limited work exists in this setting. Proposed methods include data augmentation (Volpi et al., 2018; Qiao et al., 2020; Geirhos et al., 2019), domain randomization (Tobin et al., 2017; Yue et al., 2019), self-supervised learning (Carlucci et al., 2019), and penalizing the predictive power of low-level network features (Wang et al., 2019a). Following the spirit of adding inductive bias towards global shape information over local textures, we propose using random convolutions to improve the robustness to domain shifts and small perturbations. While recently Lee et al. (2020) proposed a similar technique for improving the generalization of reinforcement learning agents in

---

[1]Code is available at `https://github.com/wildphoton/RandConv`.

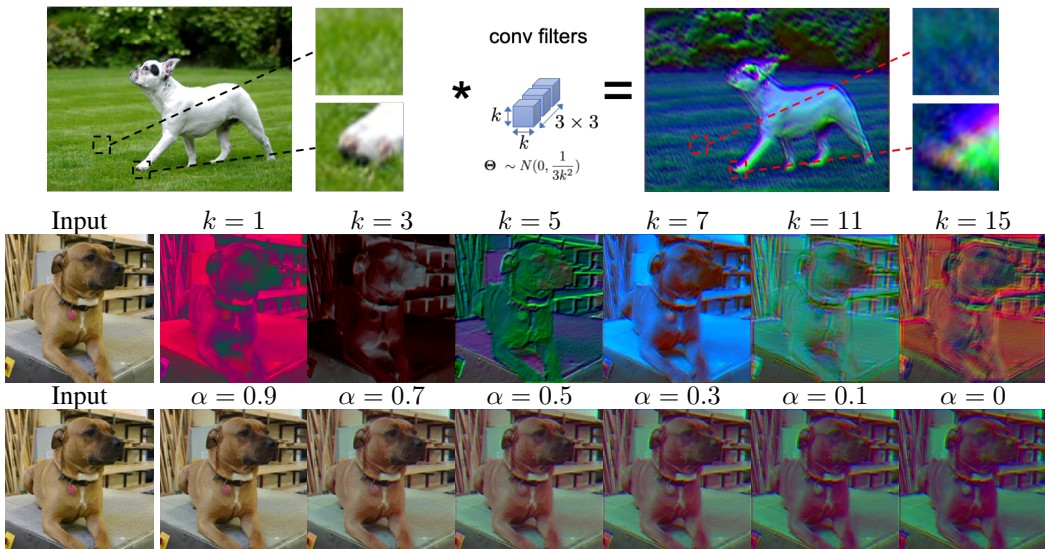

Figure 1: **Top**: Illustration that `RandConv` randomize local texture but preserve shapes in the image. **Middle:** First column is the input image of size $224^2$; following columns are convolutions results using random filters of different sizes $k$. **Bottom:** Mixing results between an image and one of its random convolution results with different mixing coefficients $\alpha$.

unseen environments, we focus on visual representation learning and examine our approach on visual domain generalization benchmarks. Our method also includes the multiscale design and a mixing variant. In addition, considering that many computer vision tasks rely on training deep networks based on ImageNet-pretrained weights (including some domain generalization benchmarks), we ask *"Can a more robust pretrained model make the finetuned model more robust on downstream tasks?"* Different from (Kornblith et al., 2019; Salman et al., 2020) who studied the transferability of a pretrained ImageNet representation to new tasks while focusing on in-domain generalization, we explore generalization performance on *unseen domains* for new tasks.

We make the following contributions:

- We develop `RandConv`, a data augmentation technique *using multi-scale random-convolutions to generate images with random texture while maintaining global shapes.* We explore using the `RandConv` output as training images or mixing it with the original images. We show that a consistency loss can further enforce invariance under texture changes.

- We provide insights and justification on why `RandConv` augments images with different local texture but the same semantics with the shape-preserving property of random convolutions.

- We validate `RandConv` and its mixing variant in extensive experiments on synthetic and real-world benchmarks as well as on the large-scale ImageNet dataset. Our methods outperform single domain generalization approaches by a large margin on digit recognition datasets and for the challenging case of generalizing to the Sketch domain in PACS and to ImageNet-Sketch.

- We explore if the robustness/generalizability of a pretrained representation can transfer. We show that transferring a model pretrained with `RandConv` on ImageNet can further improve domain generalization performance on new downstream tasks on the PACS dataset.

## 2   RELATED WORK

**Domain Generalization** (DG) aims at learning representations that perform well when transferred to unseen domains. Modern techniques range between feature fusion (Shen et al., 2019), meta-learning (Li et al., 2018a; Balaji et al., 2018), and adversarial training (Shao et al., 2019; Li et al., 2018b). Note that most current DG work (Ghifary et al., 2016; Li et al., 2018a;b) requires a multi-source training setting to work well. However, in practice, it might be difficult and expensive to collect data from multiple sources, such as collecting data from multiple medical centers (Raghupathi & Raghupathi, 2014). Instead, we consider the more strict single-domain generalization DG setting,

where we train the model on source data from a single domain and generalize it to new unseen domains (Carlucci et al., 2019; Wang et al., 2019b).

**Domain Randomization** (DR) was first introduced as a DG technique by Tobin et al. (2017) to handle the domain gap between simulated and real data. As the training data in (Tobin et al., 2017) is synthesized in a virtual environment, it is possible to generate diverse training samples by randomly selecting background images, colors, lighting, and textures of foreground objects. When a simulation environment is not accessible, image stylization can be used to generate new domains (Yue et al., 2019; Geirhos et al., 2019). However, this requires extra effort to collect data and to train an additional model; further, the number of randomized domains is limited by the number of predefined styles.

**Data Augmentation** has been widely used to improve the generalization of machine learning models (Simard et al., 2003). DR approaches can be considered a type of synthetic data augmentation. To improve performance on unseen domains, Volpi et al. (2018) generate adversarial examples to augment the training data; Qiao et al. (2020) extend this approach via meta-learning. As with other adversarial training algorithms, significant extra computation is required to obtain adversarial examples.

**Learning Representations Biased towards Global Shape** Geirhos et al. (2019) demonstrated that convolutional neural networks (CNNs) tend to use superficial local features even when trained on large datasets. To counteract this effect, they proposed to train on stylized ImageNet, thereby forcing a network to rely on object shape instead of textures. Wang et al. improved out-of-domain performance by penalizing the correlation between a learned representation and superficial features such as the gray-level co-occurrence matrix (Wang et al., 2019b), or by penalizing the predictive power of local, low-level layer features in a neural network via an adversarial classifier (Wang et al., 2019a). Our approach shares the idea that learning representations invariant to local texture helps generalization to unseen domains. However, `RandConv` avoids searching over many hyper-parameters, collecting extra data, and training other networks. It also scales to large-scale datasets since it adds minimal computation overhead.

**Random Mapping in Machine Learning** Random projections have also been effective for dimensionality reduction based on the distance-preserving property of the Johnson–Lindenstrauss lemma (Johnson & Lindenstrauss, 1984). (Vinh et al., 2016) applied random projections on entire images as data augmentation to make neural networks robust to adversarial examples. Lee et al. (2020) recently used random convolutions to help reinforcement learning (RL) agents generalize to new environments. Neural networks with *fixed* random weights can encode meaningful representations (Saxe et al., 2011) and are therefore useful for neural architecture search (Gaier & Ha, 2019), generative models (He et al., 2016b), natural language processing (Wieting & Kiela, 2019), and RL (Osband et al., 2018; Burda et al., 2019). In contrast, `RandConv` uses *non-fixed* randomly-sampled weights to generate images with different local texture.

# 3 RANDCONV: RANDOMIZE LOCAL TEXTURE AT DIFFERENT SCALES

We propose using a convolution layer with non-fixed random weights as the first layer of a DNN during training. This strategy generates images with random local texture but consistent shapes, and is beneficial for robust visual representation learning. Sec. 3.1 justifies the shape-preserving property of a random convolution layer. Sec. 3.2 describes `RandConv`, our data augmentation algorithm using a multi-scale randomized convolution layer and input mixing.

## 3.1 A RANDOM CONVOLUTION LAYER PRESERVES GLOBAL SHAPES

Convolution is the key building block for deep convolutional neural networks. Consider a convolution layer with filters $\Theta \in \mathbb{R}^{h \times w \times C_{in} \times C_{out}}$ with an input image $\mathbf{I} \in \mathbb{R}^{H \times W \times C_{in}}$, where $H$ and $W$ are the height and width of the input and $C_{in}$ and $C_{out}$ are the number of feature channels for the input and output, and $h$ and $w$ are the height and width of the layer's filter. The output (with appropriate input padding) will be $\mathbf{g} = \mathbf{I} * \Theta$ with $\mathbf{g} \in \mathbb{R}^{H \times W \times C_{out}}$.

In images, nearby pixels with similar color or texture can be grouped into primitive shapes that represent parts of objects or the background. A convolution layer linearly projects local image patches to features at corresponding locations on the output map using shared parameters. While a

convolution with random filters can project local patches to arbitrary output features, the output of a random linear projection approximately preserves relative similarity between input patches, proved in Appendix B. In other words, since any two locations within the same shape have similar local textures in the input image, they tend to be similar in the output feature map. Therefore, shapes that emerge in the output feature map are similar to shapes in the input image provided that the filter size is sufficiently small compared to the size of a typical shape.

In other words, the size of a convolution filter determines the smallest shape it can preserve. For example, 1x1 random convolutions preserve shapes at the single-pixel level and thus work as a random color mapping; large filters perturb shapes smaller than the filter size that are considered local texture of a shape at this larger scale. See Fig. 1 for examples. *More discussion and a formal proof are in Appendix A and B.*

### 3.2 Multi-scale Image Augmentation with a Randomized Convolution Layer

---

**Algorithm 1** Learning with Data Augmentation by Random Convolutions

---

1: **Input**: Model $\Phi$, task loss $\mathcal{L}_{task}$, training images $\{I_i\}_{i=1}^N$ and their labels $\{y_i\}_{i=1}^N$, pool of filter sizes $\mathcal{K} = \{1, ..., n\}$, fraction of original data $p$, whether to `mix` with original images, consistency loss weight $\lambda$
2: **function** RANDCONV(I, $\mathcal{K}$, `mix`, $p$)
3:      Sample $p_0 \sim U(0, 1)$
4:      **if** $p_0 < p$ and `mix` is False **then**
5:          return $I$                           ▷ When not in `mix` mode, use the original image with probability $p$
6:      **else**
7:          Sample scale $k \sim \mathcal{K}$
8:          Sample convolution weights $\Theta \in \mathbb{R}^{k \times k \times 3 \times 3} \sim N(0, \frac{1}{3k^2})$
9:          $I_{rc} = I * \Theta$                                       ▷ Apply convolution on $I$
10:          **if** `mix` is True **then**
11:              Sample $\alpha \sim U(0, 1)$
12:              return $\alpha I + (1 - \alpha) I_{rc}$                      ▷ Mix with original images
13:          **else**
14:              return $I_{rc}$
15: **Learning Objective**:
16: **for** $i = 1 \rightarrow N$ **do**
17:      **for** $j = 1 \rightarrow 3$ **do**
18:          $\hat{y}_i^j = \Phi(\text{RandConv}(I_i))$          ▷ Predict labels for three augmented variants of the same image
19:      $\mathcal{L}_{cons} = \lambda \sum_{j=1}^3 \text{KL}(\hat{y}_i^j || \bar{y}_i)$ where $\bar{y}_i = \sum_{j=1}^3 \hat{y}_i^j / 3$          ▷ Consistency Loss
20:      $\mathcal{L} = \mathcal{L}_{task}(\hat{y}_i^1, y_i) + \lambda \mathcal{L}_{cons}$          ▷ Learning with the task loss and the consistency loss

---

Sec. 3.1 discussed how outputs of randomized convolution layers approximately maintain shape information at a scale larger than their filter sizes. Here, we develop our `RandConv` data augmentation technique using a randomized convolution layer with $C_{out} = C_{in}$ to generate shape-consistent images with randomized texture (see Alg. 1). Our goal is not to use `RandConv` to parameterize or represent texture as in previous filter-bank based texture models (Heeger & Bergen, 1995; Portilla & Simoncelli, 2000). Instead, we only use the three-channel outputs of `RandConv` as new images with the same shape and different "style" (loosely referred to as "texture"). We also note that, a convolution layer is different from a convolution operation in image filtering. Standard image filtering applies the same 2D filter on three color channels separately. In contrast, our convolution layer applies three different *3D* filters and each takes all color channels as input and generates one channel of the output. Our proposed `RandConv` variants are as follows:

**RC$_{\text{img}}$: Augmenting Images with Random Texture** A simple approach is to use the randomized convolution layer outputs, $I * \Theta$, as new images; where $\Theta$ are the randomly sampled weights and $I$ is a training image. If the original training data is in the domain $D^0$, a sampled weight $\Theta_k$ generates images with consistent global shape but random texture forming the random domain $D^k$. Thus, by random weight sampling, we obtain an infinite number of random domains $D^1, D^1, \ldots, D^\infty$. Input image intensities are assumed to be a standard normal distribution $N(0, 1)$ (which is often true in practice thanks to data whitening). As the outputs of `RandConv` should follow the same distribution, we sample the convolution weights from $N(0, \sigma^2)$ where $\sigma = 1/\sqrt{C_{in} \times h \times w}$, which is commonly applied for network initialization (He et al., 2015). We include the original images for training at a ratio $p$ as a hyperparameter.

**RC$_{\text{mix}}$: Mixing Variant** As shown in Fig. 1, outputs from RC$_{\text{img}}$ can vary significantly from the appearance of the original images. Although generalizing to domains with significantly different local texture distributions is useful, we may not want to sacrifice much performance on domains similar to the training domain. Inspired by the AugMix (Hendrycks et al., 2020b) strategy, we propose to blend the original image with the outputs of the `RandConv` layer via linear convex combinations $\alpha I + (1 - \alpha)(I * \Theta)$, where $\alpha$ is the mixing weight uniformly sampled from $[0, 1]$. In RC$_{\text{mix}}$, the `RandConv` outputs provide shape-consistent perturbations of the original images. Varying $\alpha$, we continuously interpolate between the training domain and the randomly sampled domains of RC$_{\text{img}}$.

**Multi-scale Texture Corruption** As discussed in Sec. 3.1,, image shape information at a scale smaller than a filter's size will be corrupted by `RandConv`. Therefore, we can use filters of varying sizes to preserve shapes at various scales. We choose to uniformly randomly sample a filter size $k$ from a pool $\mathcal{K} = 1, 3, ...n$ before sampling convolution weights $\Theta \in \mathbb{R}^{k \times k \times C_{in} \times C_{out}}$ from a Gaussian distribution $N(0, \frac{1}{k^2 C_{in}})$. Fig. 1 shows examples of multi-scale `RandConv` outputs.

**Consistency Regularization** To learn representations invariant to texture changes, we use a loss encouraging consistent network predictions for the same `RandConv`-augmented image for different random filter samples. Approaches for transform-invariant domain randomization (Yue et al., 2019), data augmentation (Hendrycks et al., 2020b), and semi-supervised learning (Berthelot et al., 2019) use similar strategies. We use Kullback-Leibler (KL) divergence to measure consistency. However, enforcing prediction similarity of two augmented variants may be too strong. Instead, following (Hendrycks et al., 2020b), we use `RandConv` to obtain 3 augmentation samples of image $I$: $G_j = $ `RandConv`$^j(I)$ for $j = 1, 2, 3$ and obtain their predictions with a model $\Phi$: $y^j = \Phi(G^j)$. We then compute the *relaxed* loss as $\lambda \sum_{j=1}^{3} \text{KL}(y^j || \bar{y})$, where $\bar{y} = \sum_{j=1}^{3} y^j / 3$ is the sample average.

## 4 EXPERIMENTS

Secs. 4.1 to 4.3 evaluate our methods on the following datasets: multiple digit recognition datasets, PACS, and ImageNet-sketch. Sec. 4.4 uses PACS to explore the out-of-domain generalization of a pretrained representation in transfer learning by checking if pretraining on ImageNet with our method improves the domain generalization performance in downstream tasks. All experiments are in the single-domain generalization setting where training and validation sets are drawn from one domain. *Additional experiments with ResNet18 as the backbone are given in the Appendix.*

### 4.1 DIGIT RECOGNITION

The five digit recognition datasets (MNIST (LeCun et al., 1998), MNIST-M (Ganin et al., 2016), SVHN (Netzer et al., 2011), SYNTH (Ganin & Lempitsky, 2014) and USPS (Denker et al., 1989)) have been widely used for domain adaptation and generalization research (Peng et al., 2019a;b; Qiao et al., 2020). Following the setups in (Volpi et al., 2018) and (Qiao et al., 2020), we train a simple CNN with *10,000* MNIST samples and evaluate the accuracy on the test sets of the other four datasets. We also test on MNIST-C (Mu & Gilmer, 2019), a robustness benchmark with *15 common corruptions* of MNIST and report the average accuracy over all corruptions.

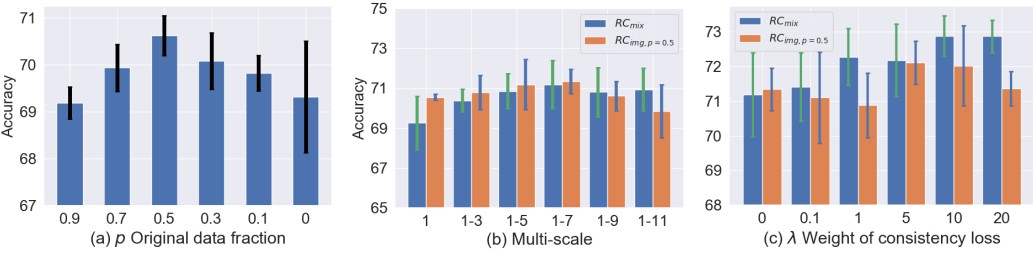

Figure 2: Average accuracy and 5-run variance of MNIST model on MNIST-M, SVHN, SYNTH and USPS. Studies for: (a) original data fraction $p$ for RC$_{\text{img}}$; (b) multiscale design (1-n refers to using scales 1,3,..,n) for RC$_{\text{img},p=0.5}$ (orange) and RC$_{\text{mix}}$ (blue); (c) consistency loss weight $\lambda$ for RC$_{\text{img}1-7,p=0.5}$ (orange) and RC$_{\text{mix}1-7}$ (blue).

**Selecting Hyperparameters and Ablation Study.** Fig. 2(a) shows the effect of the hyperparameter $p$ on $RC_{img}$ with filter size 1. We see that adding only $10\%$ `RandConv` data ($p = 0.9$) immediately improves the average performance (DG-Avg) on MNIST-M, SVHN, SYNTH and USPS performance from 53.53 to 69.19, outperforming all other approaches (see Tab. 1) for every dataset. We choose $p = 0.5$, which obtains the best DG-Avg. Fig. 2(b) shows results for a multiscale ablation study. Increasing the pool of filter sizes up to 7 improves DG-Avg performance. Therefore we use multi-scale 1-7 to study the consistency loss weight $\lambda$, shown in Fig. 2(c). Adding the consistency loss improves both `RandConv` variants on DG-avg: $RC_{mix1-7}$ favors $\lambda = 10$ while $RC_{img1-7,p=0.5}$ performs similarly for $\lambda = 5$ and $\lambda = 10$. We choose $\lambda = 10$ for all subsequent experiments.

**Results.** Tab. 1 compares the performance of $RC_{img1-7,p=0.5,\lambda=10}$ and $RC_{mix1-7,\lambda=10}$ with other state-of-the-art approaches. We show results of the adversarial training based methods GUD (Volpi et al., 2018), M-ADA (Qiao et al., 2020), and PAR (Wang et al., 2019a). The baseline model is trained only on the standard classification loss. To show `RandConv` is more than a trivial color/contrast adjustment method, we also compare to ColorJitter[2] data augmentation (which randomly changes image brightness, contrast, and saturation) and GreyScale (where images are transformed to grey-scale for training and testing). We also tested data augmentation with a fixed Laplacian of Gaussian filter (Band-Pass) of size=3 and $\sigma = 1$ and the data augmentation pipeline (Multi-Aug) that was used in a recently proposed large scale study on domain generalization algorithms and datasets (Gulrajani & Lopez-Paz, 2020). `RandConv` and its mixing variant outperforms the best competing method (M-ADA) by 17% on DG-Avg and achieves the best 91.62% accuracy on MNIST-C. While the difference between the two variants of `RandConv` is marginal, $RC_{mix1-7,\lambda=10}$ performs better on both DG-Avg and MNIST-C. When combined with Multi-Aug, `RandConv` achieves improved performance except on MNIST-C. Fig 3 shows t-SNE image feature plots for unseen domains generated by the baseline approach and $RC_{mix1-7,\lambda=10}$. The `RandConv` embeddings suggest better generalization to unseen domains.

Table 1: Average accuracy and 5-run standard deviation (in parenthesis) of MNIST10K model on MNIST-M, SVHN, SYNTH, USPS and their average (DG-avg); and average accuracy of 15 types of corruptions in MNIST-C. Both `RandConv` variants significantly outperform all other methods.

| | MNIST | MNIST-M | SVHN | USPS | SYNTH | DG-Avg | MNIST-C |
|---|---|---|---|---|---|---|---|
| Baseline | 98.40(0.84) | 58.87(3.73) | 33.41(5.28) | 79.27(2.70) | 42.43(5.46) | 53.50(4.23) | 88.20(2.10) |
| GreyScale | 98.82(0.02) | 58.41(0.99) | 36.06(1.48) | 80.45(1.00) | 45.00(0.80) | 54.98(0.86) | 89.15(0.44) |
| ColorJitter | 98.72(0.05) | 62.72(0.66) | 39.61(0.88) | 79.18(0.60) | 46.40(0.34) | 56.98(0.39) | 89.48(0.18) |
| BandPass | 98.65(0.11) | 70.22(2.73) | 48.34(2.56) | 78.60(0.82) | 57.17(2.01) | 63.58(1.89) | 87.89(0.68) |
| MultiAug | 98.80(0.05) | 62.32(0.66) | 39.07(0.68) | 79.31(1.02) | 46.48(0.80) | 56.79(0.34) | 89.54(0.11) |
| PAR (our imp) | 98.79(0.05) | 61.16(0.21) | 36.08(1.27) | 79.95(1.18) | 45.48(0.35) | 55.67(0.33) | 89.34(0.45) |
| GUD | - | 60.41 | 35.51 | 77.26 | 45.32 | 54.62 | - |
| M-ADA | - | 67.94 | 42.55 | 78.53 | 48.95 | 59.49 | - |
| $RC_{img1-7,\ p=0.5,\ \lambda=5}$ | 98.86(0.05) | 87.67(0.37) | 54.95(1.90) | 82.08(1.46) | 63.37(1.58) | 72.02(1.15) | 90.94(0.51) |
| $RC_{mix1-7,\lambda=10}$ | 98.85(0.04) | 87.76(0.83) | 57.52(2.09) | 83.36(0.96) | 62.88(0.78) | 72.88(0.58) | **91.62(0.77)** |
| $RC_{mix1-7,\lambda=10}$ + MultiAug | 98.82(0.06) | **87.89(0.29)** | **62.07(0.62)** | **84.39(1.02)** | **63.90(0.63)** | **74.56(0.46)** | 91.40(0.93) |

## 4.2 PACS Experiments

The PACS dataset (Li et al., 2018b) considers 7-class classification on 4 domains: photo, art painting, cartoon, and sketch, with very different texture styles. Most recent domain generalization work studies the multi-source domain setting on PACS and uses domain labels of the training data. Although we follow the convention to train on 3 domains and to test on the fourth, we simply pool the data from the 3 training domains as in (Wang et al., 2019a), without using domain labels during the training.

**Baseline and State-of-the-Art**. Following (Li et al., 2017), we use Deep-All as the baseline, which finetunes an ImageNet-pretrained AlexNet on 3 domains using only the classification loss and tests on the fourth domain. We test our `RandConv` variants $RC_{img1-7,p=0.5}$ and $RC_{mix1-7}$ with and without consistency loss, and ColorJitter/GreyScale/BandPass/MultiAug data augmentation as in the digit datasets. We also implemented PAR (Wang et al., 2019a) using our baseline model. $RC_{mix1-7}$

---

[2]See PyTorch documentation for implementation details; all parameters are set to 0.5.

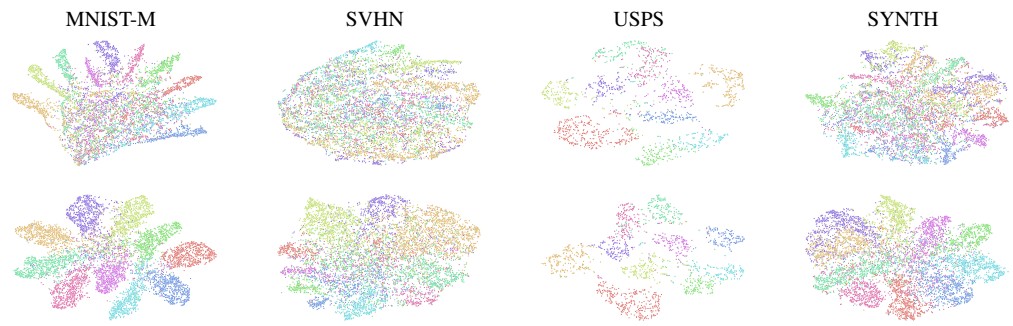

Figure 3: t-SNE feature embedding visualization for digit datasets for models trained on MNIST without (top) and with our $RC_{mix1-7, \lambda=10}$ approach (bottom). Different colors denote different classes.

Table 2: Mean and 5-run standard deviation (in parenthesis) results for domain generalization on PACS. Best results with our Deep-All baseline are in **bold**. The domain name in each column represents the target domain. Base column indicates different baselines and results under different baselines are not directly comparable. MLDG and CIDDF used domain labels for training.

| Base | Method | Photo | Art | Cartoon | Sketch | Average |
|---|---|---|---|---|---|---|
| Ours | Deep-All | 86.77(0.42) | 60.11(1.33) | 64.12(0.32) | 55.28(4.71) | 66.57(1.36) |
| | GreyScale | 83.93(1.47) | 61.60(1.18) | 62.12(0.61) | 60.07(2.47) | 66.93(0.83) |
| | ColorJitter | 84.61(0.83) | 59.01(0.24) | 61.43(0.68) | 62.44(1.68) | 66.88(0.33) |
| | BandPass | 87.08(0.57) | 59.46(0.27) | **64.39(0.51)** | 55.39(2.95) | 66.58(0.73) |
| | MultiAug | 85.21(0.47) | 59.51(0.38) | 62.88(1.01) | 61.67(0.76) | 67.32(0.23) |
| | PAR (our imp.) | **87.21(0.42)** | 60.17(0.95) | 63.63(0.88) | 55.83(2.57) | 66.71(0.58) |
| | $RC_{img1-7, p=0.5}$ | 86.50(0.72) | 61.10(0.38) | 64.24(0.62) | 68.50(1.83) | 70.09(0.43) |
| | $RC_{mix1-7}$ | 86.60(0.67) | 61.74(0.90) | 64.05(0.66) | 69.74(0.66) | **70.53(0.25)** |
| | $RC_{mix1-7}$ + MultiAug | 86.23(0.74) | **61.91(0.76)** | 62.69(0.76) | 67.74(1.21) | 69.64(0.49) |
| | $RC_{img1-7, p=0.5, \lambda=10}$ | 81.15(0.76) | 59.56(0.79) | 62.42(0.59) | 71.74(0.43) | 68.72(0.58) |
| | $RC_{mix1-7, \lambda=10}$ | 81.78(1.11) | 61.14(0.51) | 63.57(0.29) | **71.97(0.38)** | 69.62(0.24) |
| Results below are not directly comparable due to different Deep-All implementations. | | | | | | |
| Wang et al. (2019a) | Deep-All (our run) | 88.40 | 66.26 | 66.58 | 59.40 | 70.16 |
| | PAR (our run) | 88.40 | 65.19 | 68.58 | 61.86 | 71.10 |
| | PAR (reported) | 89.6 | 66.3 | 68.3 | 64.1 | 72.08 |
| Carlucci et al. (2019) | Deep-All | 89.98 | 66.68 | 69.41 | 60.02 | 71.52 |
| | Jigen | 89.00 | 67.63 | 71.71 | 65.18 | 73.38 |
| Li et al. (2018a) | Deep-All | 86.67 | 64.91 | 64.28 | 53.08 | 67.24 |
| | MLDG (use domain labels) | 88.00 | 66.23 | 66.88 | 58.96 | 70.01 |
| Li et al. (2018c) | Deep-All | 77.98 | 57.55 | 67.04 | 58.52 | 65.27 |
| | CIDDG (use domain labels) | 78.65 | 62.70 | 69.73 | 64.45 | 68.88 |

combined with MultiAug is also tested. Further, we compare to the following state-of-the-art approaches: Jigen (Carlucci et al., 2019) using self-supervision, MLDG (Li et al., 2018a) using meta-learning, and the conditional invariant deep domain generalization method CIDDG (Li et al., 2018c). Note that previous methods used different Deep-All baselines which make the final accuracy not directly comparable, and MLDG and CIDDG use domain labels for training.

**Results.** Tab. 2 shows *significant improvements on Sketch* for both `RandConv` variants. Sketch is the most challenging domain with no color and much less texture compared to the other 3 domains. The success on Sketch demonstrates that our methods can guide the DNN to learn global representations focusing on shapes that are robust to texture changes. Without using the consistency loss, $RC_{mix1-7}$ achieves the best overall result improving over Deep-All by $\sim$4% but adding MultiAug does not further improve the performance. Adding the consistency loss with $\lambda = 10$, $RC_{mix1-7}$ and

RC$_{\text{img1-7},p=0.5}$ performs better on Sketch but degrades performance on the other 3 domains, so do GreyScale and ColorJitter. *This observation will be discussed in Sec 4.4.*

### 4.3 GENERALIZING AN IMAGENET MODEL TO IMAGENET-SKETCH

Table 3: Accuracy of ImageNet-trained AlexNet on ImageNet-Sketch (IN-S) data. Our methods outperform PAR by 5% and are on par with a Stylized-ImageNet (SIN) trained model. Note that PAR was built on top of a stronger baseline than our model, and both PAR and SIN fine-tuned the baseline model which helped the performance, while we train `RandConv` model from scratch.

|  | Baseline (Wang et al., 2019a) | PAR (Wang et al., 2019a) | Baseline | RC$_{\text{img1-7},}$ $_{p=0.5,\lambda=10}$ | RC$_{\text{mix1-7},}$ $_{\lambda=10}$ | SIN (Geirhos et al., 2019) |
|---|---|---|---|---|---|---|
| Top1 | 12.04 | 13.06 | 10.28 | 18.09 | 16.91 | 17.62 |
| Top5 | 25.60 | 26.27 | 21.60 | 35.40 | 33.99 | 36.22 |

ImageNet-Sketch (Wang et al., 2019a) is an out-of-domain test set for models trained on ImageNet. We trained AlexNet from scratch with RC$_{\text{img1-7},p=0.5,\lambda=10}$ and RC$_{\text{mix1-7},\lambda=10}$. We evaluate their performance on ImageNet-Sketch. We use the AlexNet model trained without `RandConv` as our baseline. Tab. 3 compares `PAR` and its baseline model and AlexNet trained with Stylized ImageNet (SIN) (Geirhos et al., 2019) on ImageNet-Sketch. Although `PAR` uses a stronger baseline, `RandConv` achieves significant improvements over our baseline and outperforms `PAR` by a large margin. Our methods achieve more than a 7% accuracy improvement over the baseline and surpass PAR by 5%. SIN as an image stylization approach that can modify image texture in a hierarchical and realistic way. However, albeit its complexity, it still performs on par with RandConv. Note that image stylization techniques require additional data and heavy precomputation. Further, the images for the style source also need to be chosen. In contrast, RandConv is much easier to use: it can be applied to any dataset via a simple convolution layer. We also measure the shape-bias metric proposed by Geirhos et al. (2019) for `RandConv` trained AlexNet. RC$_{\text{img1-7},p=0.5,\lambda=10}$ and RC$_{\text{mix1-7},\lambda=10}$ improve the baseline from 25.36% to 48.24% and 54.85% respectively.

### 4.4 REVISITING PACS WITH MORE ROBUST PRETRAINED REPRESENTATIONS

A common practice for many computer vision tasks (including the PACS benchmark) is transfer learning, i.e. finetuning a backbone model pretrained on ImageNet. Recently, how the accuracy on ImageNet (Kornblith et al., 2019) and adversarial robustness (Salman et al., 2020) of the pretrained model affect transfer learning has been studied in the context of domain generalization. Instead, we study how out-of-domain generalizability transfers from pretraining to downstream tasks and shed light on how to better use pretrained models.

**Impact of ImageNet Pretraining** A model trained on ImageNet may be biased towards textures (Geirhos et al., 2019). Finetuning ImageNet pretrained models on PACS may inherit this texture bias, thereby benefitting generalization on the Photo domain (which is similar to ImageNet), but hurting performance on the Sketch domain. Therefore, as shown in Sec. 4.2, using `RandConv` to correct this texture bias improves results on Sketch, but degrades them on the Photo domain. Since pretraining has such a strong impact on transfer performance to new tasks, we ask: *"Can the generalizability of a pretrained model transfer to downstream tasks? I.e., does a pretrained model with better generalizability improve performance on unseen domains on new tasks?"* To answer this, we revisit the PACS tasks based on ImageNet-pretrained weights where our two `RandConv` variants of Sec. 4.3 are used during ImageNet training. We study if this results in performance changes for the Deep-All baseline and for finetuning with `RandConv`.

**Better Performance via RandConv pretrained model** We start by testing the Deep-All baselines using the two `RandConv`-trained ImageNet models of Sec. 4.3 as initialization. Tab. 4 shows significant improvements on Sketch. Results are comparable to finetuning with `RandConv` on a normal pretrained model. Art is also consistently improved. Performance drops slightly on Photo as expected, since we reduced the texture bias in the pretrained model, which is helpful for the Photo domain. A similar performance improvement is observed when using the SIN-trained AlexNet as initialization. Using `RandConv` for *both* ImageNet training and PACS finetuning, we achieve 76.11% accuracy on Sketch. As far as we know, this is the best performance using an AlexNet baseline. This approach even outperforms Jigen (Carlucci et al., 2019) (71.35%) with a stronger ResNet18 baseline

Table 4: Generalization results on PACS with `RandConv` and SIN pretrained AlexNet. ImageNet column shows how the pretrained model is trained on ImageNet (baseline represents training the ImageNet model using only the classification loss); PACS column indicates the methods used for finetuning on PACS. **Best** and second best accuracy for each target domain are highlighted in bold and underlined.

| PACS | ImageNet | Photo | Art | Cartoon | Sketch | Avg |
|---|---|---|---|---|---|---|
| | Baseline | **86.77**$_{(0.42)}$ | 60.11$_{(1.33)}$ | 64.12$_{(0.32)}$ | 55.28$_{(4.71)}$ | 66.57$_{(1.36)}$ |
| Deep-All | RC$_{img1\text{-}7,p=0.5,\lambda=10}$ | 84.48$_{(0.52)}$ | 62.61$_{(1.23)}$ | 66.13$_{(0.80)}$ | 69.24$_{(0.80)}$ | 70.61$_{(0.53)}$ |
| | RC$_{mix1\text{-}7,\lambda=10}$ | 85.59$_{(0.40)}$ | 63.30$_{(0.99)}$ | 63.83$_{(0.85)}$ | 68.29$_{(1.27)}$ | 70.25$_{(0.45)}$ |
| | SIN | 85.33$_{(0.66)}$ | **65.85**$_{(0.87)}$ | 65.39$_{(0.62)}$ | 65.75$_{(0.59)}$ | 70.58$_{(0.21)}$ |
| RC$_{img1\text{-}7,}$ | Baseline | 81.15$_{(0.76)}$ | 59.56$_{(0.79)}$ | 62.42$_{(0.59)}$ | 71.74$_{(0.43)}$ | 68.72$_{(0.58)}$ |
| $p=0.5,\lambda=10$ | RC$_{img1\text{-}7,p=0.5,\lambda=10}$ | 84.36$_{(0.36)}$ | 63.73$_{(0.91)}$ | **68.07**$_{(0.55)}$ | 75.41$_{(0.57)}$ | 72.89$_{(0.33)}$ |
| | RC$_{mix1\text{-}7,\lambda=10}$ | 84.63$_{(0.97)}$ | 63.41$_{(1.22)}$ | 66.36$_{(0.43)}$ | 74.59$_{(0.84)}$ | 72.25$_{(0.54)}$ |
| RC$_{mix1\text{-}7}$ | Baseline | 81.78$_{(1.11)}$ | 61.14$_{(0.51)}$ | 63.57$_{(0.29)}$ | 71.97$_{(0.38)}$ | 69.62$_{(0.24)}$ |
| $\lambda=10$ | RC$_{img1\text{-}7,p=0.5,\lambda=10}$ | 85.16$_{(1.03)}$ | 63.17$_{(0.38)}$ | 67.68$_{(0.60)}$ | **76.11**$_{(0.43)}$ | **73.03**$_{(0.46)}$ |
| | RC$_{mix1\text{-}7,\lambda=10}$ | 86.17$_{(0.56)}$ | 65.33$_{(1.05)}$ | 65.52$_{(1.13)}$ | 73.21$_{(1.03)}$ | 72.56$_{(0.50)}$ |

model. Cartoon and Art are also improved. The best average domain generalization accuracy is 73.03%, with a more than 6% improvement over our initial Deep-All baseline.

This experiment confirms that generalizability may transfer: removing texture bias may not only make a pretrained model more generalizable, but it may help generalization on downstream tasks. For similar target and pretraining domains like Photo and ImageNet, where learning texture bias may actually be beneficial, performance may degrade slightly.

## 5 CONCLUSION AND DISCUSSION

Randomized convolution (`RandConv`) is a simple but powerful data augmentation technique for randomizing local image texture. `RandConv` helps focus visual representations on global shape information rather than local texture. We theoretically justified the approximate shape-preserving property of `RandConv` and developed `RandConv` techniques using multi-scale and mixing designs. We also make use of a consistency loss to encourage texture invariance. `RandConv` outperforms state-of-the-art approaches on the digit recognition benchmark and on the sketch domain of PACS and on ImageNet-Sketch by a large margin. By finetuning a model pretrained with `RandConv` on PACS, we showed that the generalizability of a pretrained model may transfer to and benefit a new downstream task. This resulted in a new state-of-art performance on PACS in the Sketch domain.

`RandConv` can help computer vision tasks when a shape-biased model is helpful e.g. for object detection. `RandConv` can also provide a shape-biased pretrained model to improve performance on downstream tasks when generalizing to unseen domains. However, local texture features can be useful for many computer vision tasks, especially for fixed-domain fine-grained visual recognition. In such cases, visual representations that are invariant to local texture may hurt in-domain performance. Therefore, important future work includes learning representations that disentangle shape and texture features and building models to use such representations in an explainable way.

Adversarial robustness of deep neural networks has received significant recent attention. Interestingly, Zhang & Zhu (2019) find that adversarially-trained models are more shape biased; Shi et al. (2020) show that their method for increasing shape bias also helps adversarial robustness, especially when combined with adversarial training. Therefore, exploring how `RandConv` affects the adversarial robustness of models could be interesting future work. Moreover, recent biologically inspired models for improving adversarial robustness (Dapello et al., 2020) use Gabor filters with fixed random configurations followed by a stochastic layer to add Gaussian noise to the network input, which may explain the importance of randomness in `RandConv`. Exploring connections between `RandConv` and biological mechanisms in the human visual system would be interesting future work.

**Acknowledgments** We thank Zhiding Yu for discussions on initial ideas and the experimental setup. We also thank Nathan Cahill for advice on proving the properties of random convolutions.

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

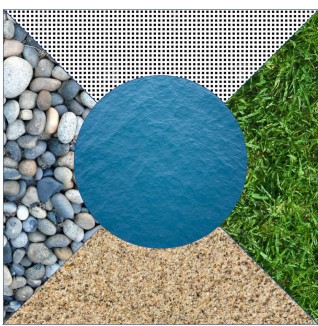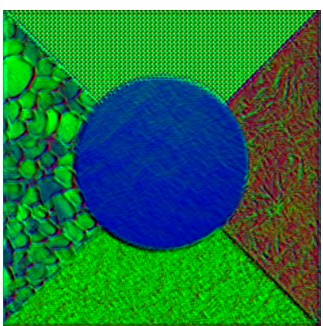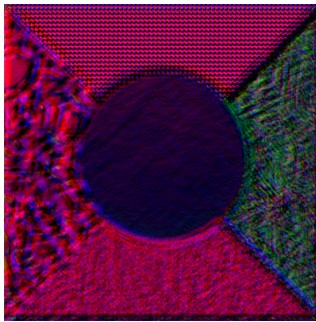

Figure 4: **Left:** An image with texture and shapes at different scales; **Middle:** The output of `RandConv` with a small filter size which largely preserves the shapes of the stones. **Right:** The output of `RandConv` with a large filter size distorts the shape of the stones as well.

This supplementary material provides additional details. Specifically, in Sec. A and B, we discuss definitions of shapes and textures in images and justify why random convolution preserves global shapes and disrupts local texture formally by proving Theorem 1. This theorem shows that random linear projections are approximately distance preserving. We also discuss our simulation-based bound based on 80% distance rescaling on real image data. Sec. C provides more experimental details for the different datasets. Sec. D shows experimental results with a stronger backbone architecture and on a new benchmark ImageNet-R (Hendrycks et al., 2020a). Sec. E provides more detailed results regarding hyperparameter selection and ablation studies. Lastly, Sec. F shows example visualizations of `RandConv` outputs and for its mixing variant.

## A    SHAPES AND TEXTURE IN IMAGES

As discussed in the main text, we define shapes in images that are preserved by a random convolution layer as primitive shapes: spatial clusters of pixels with similar local texture. An object in a image can be a single primitive shape alone but in most cases it is the composition of multiple primitive shapes e.g. a car includes wheels, body frames, windshields. Note that the definition of texture is not necessarily opposite to shapes, since the texture of a larger shape can includes smaller shapes. For example, in Fig.4, the left occluded triangle shape has texture composed by shapes of cobble stones while cobble stones have their own texture. Random convolution can preserve those large shapes that usually define the image semantics while distorting the small shapes as local texture.

To formally define the shape-preserving property, we assume $(x_1, y_1)$, $(x_2, y_2)$ and $(x_3, y_3)$ are three locations on a image and $(x_1, y_1)$ has closer color and local texture with $(x_2, y_2)$ than $(x_3, y_3)$. For example, $(x_1, y_1)$ and $(x_2, y_2)$ are within the same shape while $(x_3, y_3)$ is located at a neighboring shape. Then we have $\|\mathbf{p}(x_1, y_1) - \mathbf{p}(x_2, y_2)\| < \|\mathbf{p}(x_1, y_1) - \mathbf{p}(x_3, y_3)\|$, where $\mathbf{p}(x_i, y_i)$ is the image patch at location $(x_i, y_i)$. A transformation $f$ is *shape-preserving* if it *maintains* such relative distance relations for most location triplets, i.e.

$$\|f(\mathbf{p}(x_i, y_i)) - f(\mathbf{p}(x_j, y_j))\| / \|\mathbf{p}(x_i, y_i) - \mathbf{p}(x_j, y_j)\| \approx r \tag{1}$$

for any two spatial location $(x_i, y_i)$ and $(x_j, y_j)$; $r \geq 0$ is a constant.

## B    RANDOM CONVOLUTION IS SHAPE-PRESERVING AS RANDOM LINEAR PROJECTION IS DISTANCE PRESERVING

We can express a convolution layer as a local linear projection:

$$\mathbf{g}(x, y) = \mathbf{U}\mathbf{p}(x, y), \tag{2}$$

where $\mathbf{p}(x, y) \in \mathbb{R}^d$ ($d = h \times w \times C_{in}$) is the vectorized image patch centerized at location $(x, y)$, $\mathbf{g}(x, y) \in \mathbb{R}^{C_{out}}$ is the output feature at location $(x, y)$, and $\mathbf{U} \in \mathbb{R}^{C_{out} \times d}$ is the matrix expressing the convolution layer filters $\mathbf{\Theta}$. I.e., for each sliding window centered at $(x, y)$, a convolution layer applies a linear transform $f : \mathbb{R}^d \to \mathbb{R}^{C_{out}}$ projecting the $d$ dimensional local image patch $\mathbf{p}(x, y)$ to its $C_{out}$ dimensional feature $\mathbf{g}(x, y)$. When $\mathbf{\Theta}$ is independently randomly sampled, e.g. from

a Gaussian distribution, the convolution layer preserves global shapes since that a random linear projection is *approximately* distance-preserving by bounding the range of $r$ in Eq. 1 in Theorem 1.

**Theorem 1.** *Suppose we have $N$ data points $\mathbf{z}_1, \cdots, \mathbf{z}_N \in \mathbb{R}^d$. Let $f(\mathbf{z}) = \mathbf{U}\mathbf{z}$ be a random linear projection $f : \mathbb{R}^d \to \mathbb{R}^m$ such that $\mathbf{U} \in \mathbb{R}^{m \times d}$ and $\mathbf{U}_{i,j} \sim N(0, \sigma^2)$. Then we have:*

$$P\Big(\sup_{i \neq j; i,j \in [N]} \Big\{r_{i,j} := \frac{\|f(\mathbf{z}_i) - f(\mathbf{z}_j)\|}{\|\mathbf{z}_i - \mathbf{z}_j\|}\Big\} > \delta_1\Big) \leq \epsilon,$$

$$P\Big(\inf_{i \neq j; i,j \in [N]} \Big\{r_{i,j} := \frac{\|f(\mathbf{z}_i) - f(\mathbf{z}_j)\|}{\|\mathbf{z}_i - \mathbf{z}_j\|}\Big\} < \delta_2\Big) \leq \epsilon,$$

(3)

*where $\delta_1 := \sigma\sqrt{\chi^2_{\frac{2\epsilon}{N(N-1)}}(m)}$ and $\delta_2 := \sigma\sqrt{\chi^2_{1-\frac{2\epsilon}{N(N-1)}}(m)}$. Here, $\chi^2_\alpha(m)$ denotes the $\alpha$-upper quantile of the $\chi^2$ distribution with $m$ degrees of freedom.*

Thm. 1 tells us that for any data pair $(\mathbf{z}_i, \mathbf{z}_j)$ in a set of $N$ points, the distance rescaling ratio $r_{i,j}$ after a random linear projection is bounded by $\delta_1$ and $\delta_2$ with probability $1 - \epsilon$. A Smaller $N$ and a larger output dimension $m$ give better bounds. E.g., when $m = 3$, $N = 1,000$, $\sigma = 1$ and $\epsilon = 0.1$, $\delta_1 = 5.8$ and $\delta_2 = 0.01$. Thm. 1 gives a theoretical bound for *all* the $N(N-1)/2$ pairs. However, in practice, preserving distances for a majority of $N(N-1)/2$ pairs is sufficient. To empirically verify this, we test the range of central 80% of $\{r_{i,j}\}$ on real image data. Using the same $(m, N, \sigma, \epsilon)$, 80% of the pairs lie in $[0.56, 2.87]$, which is significantly better than the strict bound: $[0.01, 5.8]$. A proof of the theorem and simulation details are given in the following.

*Proof.* Let $\mathbf{U}_k$ represent to the $k$-th row of $\mathbf{U}$. It is easy to check that $\mathbf{v}_k := \langle \mathbf{U}_k, \mathbf{z}_i - \mathbf{z}_j\rangle / \|\mathbf{z}_i - \mathbf{z}_j\| \sim N(0, \sigma^2)$. Therefore,

$$\frac{\|f(\mathbf{z}_i) - f(\mathbf{z}_j)\|^2}{\sigma^2\|\mathbf{z}_i - \mathbf{z}_j\|^2} = \frac{1}{\sigma^2}\frac{(\mathbf{z}_i - \mathbf{z}_j)^\top \mathbf{U}^\top \mathbf{U}(\mathbf{z}_i - \mathbf{z}_j)}{\|\mathbf{z}_i - \mathbf{z}_j\|^2} = \sum_{k=1}^m \frac{\mathbf{v}_k^2}{\sigma^2} \sim \chi^2(m).$$

Therefore, for $0 < \epsilon < 1$, we have

$$P\Big(\frac{\|f(\mathbf{z}_i) - f(\mathbf{z}_j)\|^2}{\sigma^2\|\mathbf{z}_i - \mathbf{z}_j\|^2} > \chi^2_{\frac{2\epsilon}{N(N-1)}}(m)\Big) \leq \frac{2\epsilon}{N(N-1)}.$$

From the above inequality, we have

$$P\Big(\sup_{i \neq j; i,j \in [N]} \Big\{\frac{\|f(\mathbf{z}_i) - f(\mathbf{z}_j)\|^2}{\|\mathbf{z}_i - \mathbf{z}_j\|^2}\Big\} > \sigma^2 \chi^2_{\frac{2\epsilon}{N(N-1)}}(m)\Big)$$

$$= P\Big(\sup_{i \neq j; i,j \in [N]} \Big\{\frac{\|f(\mathbf{z}_i) - f(\mathbf{z}_j)\|^2}{\sigma^2\|\mathbf{z}_i - \mathbf{z}_j\|^2}\Big\} > \chi^2_{\frac{2\epsilon}{N(N-1)}}(m)\Big)$$

$$= P\Big(\bigcup_{i \neq j; i,j \in [N]} \Big\{\frac{\|f(\mathbf{z}_i) - f(\mathbf{z}_j)\|^2}{\sigma^2\|\mathbf{z}_i - \mathbf{z}_j\|^2} > \chi^2_{\frac{2\epsilon}{N(N-1)}}(m)\Big\}\Big)$$

$$\leq \sum_{i \neq j; i,j \in [N]} P\Big(\frac{\|f(\mathbf{z}_i) - f(\mathbf{z}_j)\|^2}{\sigma^2\|\mathbf{z}_i - \mathbf{z}_j\|^2} > \chi^2_{\frac{2\epsilon}{N(N-1)}}(m)\Big)$$

$$\leq \epsilon,$$

which is equivalent to

$$P\Big(\sup_{i \neq j; i,j \in [N]} \Big\{\frac{\|f(\mathbf{z}_i) - f(\mathbf{z}_j)\|}{\|\mathbf{z}_i - \mathbf{z}_j\|}\Big\} > \sigma\sqrt{\chi^2_{\frac{2\epsilon}{N(N-1)}}(m)}\Big) \leq \epsilon.$$

Similarly, we have

$$P\Big(\inf_{i \neq j; i,j \in [N]} \Big\{\frac{\|f(\mathbf{z}_i) - f(\mathbf{z}_j)\|}{\|\mathbf{z}_i - \mathbf{z}_j\|}\Big\} < \sigma\sqrt{\chi^2_{1-\frac{2\epsilon}{N(N-1)}}(m)}\Big) \leq \epsilon.$$

$\square$

**Simulation on Real Image Data** To better understand the relative distance preservation property of random linear projections in practice, we use Algorithm 2 to empirically obtain a bound for real image data. We choose $m = 3$, $N = 1,000$, $\sigma = 1$ and $\epsilon = 0.1$ as in computing our theoretical bounds. We use $M = 1,000$ real images from the PACS dataset for this simulation. Note that the image patch size or $d$ does not affect the bound. We use a patch size of $3 \times 3$ resulting in $d = 27$. This simulation tell us that applying linear projections with a randomly sampled $U$ on $N$ local images patches in every image, we have a $1 - \epsilon$ chance that 80% of $r_{i,j}$ is in the range $[\delta_{10\%}, \delta_{90\%}]$.

---

**Algorithm 2** Simulate the range of central 80% of $r_{i,j}$ on real image data

---

1: **Input**: $M$ images $\{I_i\}_{i=1}^M$, number of data points $N$, projection output dimension $m$, standard deviation $\sigma$ of normal distribution, confidence level $\epsilon$.
2: **for** $m = 1 \rightarrow M$ **do**
3:     Sample images patches in $I_m$ at 1,000 locations and vectorize them as $\{\mathbf{z}_l^m\}_{l=1}^N$
4:     Sample a projection matrix $\mathbf{U} \in \mathbb{R}^{m \times d}$ and $\mathbf{U}_{i,j} \sim N(0, \sigma^2)$
5:     **for** $i = 1 \rightarrow N$ **do**
6:         **for** $j = i + 1 \rightarrow N$ **do**
7:             Compute $r_{i,j}^m = \frac{\|f(\mathbf{z}_i^m) - f(\mathbf{z}_j^m)\|}{\|\mathbf{z}_i^m - \mathbf{z}_j^m\|}$, where $f(\mathbf{z}) = \mathbf{U}\mathbf{z}$
8:     $q_{10\%}^m = 10\%$ quantile of $r_{i,j}^m$ for $I_m$
9:     $q_{90\%}^m = 90\%$ quantile of $r_{i,j}^m$ for $I_m$               ▷ Get the central 80% of $r_{i,j}$ in each image
10: $\delta_{10\%} = \epsilon$ quantile of all $q_{10\%}^m$
11: $\delta_{90\%} = (1 - \epsilon)$ quantile of all $q_{90\%}^m$               ▷ Get the $\epsilon$ confident bound for $q_{10\%}^m$ and $q_{90\%}^m$
12: **return** $\delta_{10\%}, \delta_{90\%}$

---

## C EXPERIMENTAL DETAILS

**Digits Recognition** The network for our digits recognition experiments is composed of two *Conv5×5-ReLU-MaxPool2×2* blocks with 64/128 output channels and three fully connected layer with 1024/1024/10 output channels. We train the network with batch size 32 for 10,000 iterations. During training, the model is validated every 250 iterations and saved with the best validation score for testing. We apply the `Adam` optimizer with an initial learning rate of 0.0001.

**PACS** We use the official data splits for training/validation/testing; no extra data augmentation is applied. We use the official `PyTorch` implementation and the pretrained weights of AlexNet for our PACS experiments. AlextNet is finetuned for 50,000 iterations with a batch size 128. Samples are randomly selected from the training data mixed between the three domains. We use the validation data of source domains only at every 100 iterations. We use the `SGD` optimizer for training with an initial learning rate of 0.001, Nesterov momentum, and weight decay set to 0.0005. We let the learning rate decay by a factor of 0.1 after finishing 80% of the iterations.

**ImageNet** Following the `PyTorch` example [3] on training ImageNet models, we set the batch size to 256 and train AlexNet from scratch for 90 epochs. We apply the `SGD` optimizer with an initial learning rate of 0.01, momentum 0.9, and weight decay 0.0001. We reduce the learning rate via a factor of 0.1 every 30 epochs.

## D MORE EXPERIMENTS WITH RESNET-18

In this section, we demonstrate that `RandConv` also works on other stronger backbone architectures, e.g. for a Residual Network He et al. (2016a). Specifically, we run the PACS and ImageNet experiments with ResNet-18 as the baseline and `RandConv`. As Table 5 shows, `RandConv` improves the baseline using ResNet18 on ImageNet-sketch by 10.5% accuracy. When using a `RandConv` pretrained ResNet-18 on PACS, the performance of finetuning with DeepAll and `RandConv` are both improved shown in Table 7. The best average domain generalization accuracy is 84.09%, with a more than 8% improvement over our initial Deep-All baseline. A model pretrained with $\text{RC}_{\text{mix1-7},\lambda=10}$ generally performs better than when pretrained with $\text{RC}_{\text{img1-7},p=0.5,\lambda=10}$. We also provide the ResNet-18 performance of JiGen (Carlucci et al., 2019) on PACS as reference. Note

---

[3] https://github.com/pytorch/examples/tree/master/imagenet

that JiGen uses extra data augmentation and a different data split than our approach and it only improves over its own baseline by 1.5%. In addition, we test `RandConv` trained ResNet-18 on ImageNet-R (Hendrycks et al., 2020a), a domain generalization benchmark that contains images of artistic renditions of 200 object classes from the original ImageNet dataset. As Table 6 shows, `RandConv` also improve the generalization performance on ImageNet-R and reduce the gap between the in-domain (ImageNet-200) and out-of-domain (ImageNet-R) performance.

Table 5: Accuracy of ImageNet-trained ResNet-18 on ImageNet-Sketch data.

|  | Baseline | $RC_{img1-7, p=0.5, \lambda=10}$ | $RC_{mix1-7, \lambda=10}$ |
|---|---|---|---|
| Top1 | 20.23 | 28.79 | 30.70 |
| Top5 | 37.26 | 49.02 | 51.80 |

Table 6: Top 1 Accuracy of ImageNet-trained ResNet-18 on ImageNet-R data. ImageNet-200 are the original ImageNet data with the same 200 classes as ImageNet-R.

|  | Baseline | $RC_{img1-7, p=0.5, \lambda=10}$ | $RC_{mix1-7, \lambda=10}$ |
|---|---|---|---|
| ImageNet-200 (%) | 88.15 | 83.72 | 72.7 |
| ImageNet-R (%) | 33.06 | 37.38 | 35.75 |
| Gap | 55.09 | 46.34 | 36.95 |

Table 7: Generalization results on PACS with `RandConv` pretrained model using ResNet-18. ImageNet column shows how the pretrained model is trained on ImageNet (baseline represents training using only the classification loss); PACS column indicates the methods used for finetuning on PACS. **Best** and second best accuracy for each target domain are highlighted in bold and underlined. The performance of JiGen (Carlucci et al., 2019) and its baseline using ResNet-18 is also given.

| PACS | ImageNet | Photo | Art | Cartoon | Sketch | Avg |
|---|---|---|---|---|---|---|
| | Baseline | $\textbf{95.45}_{(0.43)}$ | $74.96_{(0.99)}$ | $71.48_{(1.22)}$ | $62.09_{(1.12)}$ | $76.00_{(0.37)}$ |
| Deep-All | $RC_{img1-7,p=0.5,\lambda=10}$ | $94.65_{(0.16)}$ | $73.85_{(0.97)}$ | $74.78_{(0.58)}$ | $73.51_{(1.16)}$ | $79.20_{(0.40)}$ |
| | $RC_{mix1-7,\lambda=10}$ | $94.10_{(0.43)}$ | $76.72_{(1.43)}$ | $73.41_{(1.29)}$ | $77.60_{(0.55)}$ | $80.46_{(0.74)}$ |
| | Baseline | $92.37_{(0.54)}$ | $76.50_{(0.55)}$ | $71.33_{(0.29)}$ | $79.65_{(1.32)}$ | $79.96_{(0.53)}$ |
| $RC_{img1-7, p=0.5,\lambda=10}$ | $RC_{img1-7,p=0.5,\lambda=10}$ | $94.43_{(0.22)}$ | $79.80_{(1.03)}$ | $73.40_{(0.37)}$ | $81.51_{(0.85)}$ | $82.28_{(0.38)}$ |
| | $RC_{mix1-7,\lambda=10}$ | $94.57_{(0.45)}$ | $\textbf{81.32}_{(1.00)}$ | $\textbf{76.28}_{(0.82)}$ | $\textbf{84.18}_{(0.94)}$ | $\textbf{84.09}_{(0.61)}$ |
| | Baseline | $93.57_{(0.40)}$ | $77.73_{(0.91)}$ | $71.24_{(0.91)}$ | $75.53_{(2.17)}$ | $79.52_{(0.61)}$ |
| $RC_{mix1-7 \lambda=10}$ | $RC_{img1-7,p=0.5,\lambda=10}$ | $\underline{95.23}_{(0.30)}$ | $80.56_{(0.82)}$ | $74.18_{(0.53)}$ | $80.70_{(1.43)}$ | $82.67_{(0.46)}$ |
| | $RC_{mix1-7,\lambda=10}$ | $95.01_{(0.32)}$ | $\underline{81.09}_{(1.24)}$ | $\underline{76.04}_{(0.92)}$ | $\underline{83.02}_{(0.93)}$ | $\underline{83.79}_{(0.60)}$ |
| Deep-All | Baseline | 95.73 | 77.85 | 74.86 | 67.74 | 79.05 |
| JiGen | | 96.03 | 79.42 | 75.25 | 71.35 | 80.51 |

## E HYPERPARAMETER SELECTIONS AND ABLATION STUDIES ON DIGITS RECOGNITION BENCHMARKS

We provide detailed experimental results for the digits recognition datasets. Table 8 shows results for different hyperameters $p$ for $RC_{img1}$. Table 9 shows results for an ablation study on the multi-scale design for $RC_{mix}$ and $RC_{img,p=0.5}$. Table 10 shows results for studying the consistency loss weight $\lambda$ for $RC_{mix1-7}$ and $RC_{img1-7,p=0.5}$. Tables 8, 9, and 10 correspond to Fig. 2 (a)(b)(c) in the main text respectively.

Table 8: Ablation study of hyperparameter $p$ for $RC_{img1}$ on digits recognition benchmarks. DG-Avg is the average performance on MNIST-M, SVHN, SYNTH and USPS. Best results are **bold**.

| | MNIST-10k | MNIST-M | SVHN | USPS | SYNTH | DG Avg | MNIST-C |
|---|---|---|---|---|---|---|---|
| Baseline | $98.40_{(0.84)}$ | $58.87_{(3.73)}$ | $33.41_{(5.28)}$ | $79.27_{(2.70)}$ | $42.43_{(5.46)}$ | $53.50_{(4.23)}$ | $88.20_{(2.10)}$ |
| $RC_{img1,\,p=0.9}$ | $98.68_{(0.06)}$ | $83.53_{(0.37)}$ | $53.67_{(1.54)}$ | $80.38_{(1.41)}$ | $59.19_{(0.85)}$ | $69.19_{(0.34)}$ | $\mathbf{89.79_{(0.44)}}$ |
| $RC_{img1,\,p=0.7}$ | $98.64_{(0.07)}$ | $84.17_{(0.61)}$ | $54.50_{(1.55)}$ | $\mathbf{80.85_{(0.91)}}$ | $60.25_{(0.85)}$ | $69.94_{(0.50)}$ | $89.20_{(0.60)}$ |
| $RC_{img1,\,p=0.5}$ | $98.72_{(0.08)}$ | $85.17_{(1.12)}$ | $\mathbf{55.97_{(0.54)}}$ | $80.31_{(0.85)}$ | $\mathbf{61.07_{(0.47)}}$ | $\mathbf{70.63_{(0.42)}}$ | $88.66_{(0.62)}$ |
| $RC_{img1,\,p=0.3}$ | $98.71_{(0.12)}$ | $85.45_{(0.87)}$ | $54.62_{(1.52)}$ | $79.78_{(1.40)}$ | $60.51_{(0.41)}$ | $70.09_{(0.60)}$ | $89.02_{(0.32)}$ |
| $RC_{img1,\,p=0.1}$ | $98.66_{(0.06)}$ | $85.57_{(0.79)}$ | $54.34_{(1.52)}$ | $79.21_{(0.44)}$ | $60.18_{(0.63)}$ | $69.83_{(0.38)}$ | $88.53_{(0.38)}$ |
| $RC_{img1,\,p=0}$ | $98.55_{(0.13)}$ | $\mathbf{86.27_{(0.42)}}$ | $52.48_{(3.00)}$ | $79.01_{(1.11)}$ | $59.53_{(1.14)}$ | $69.32_{(1.19)}$ | $88.01_{(0.36)}$ |

Table 9: Ablation study of multi-scale `RandConv` on digits recognition benchmarks for $RC_{mix}$ and $RC_{img,p=0.5}$. Best entries for each variant are **bold**.

| | MNIST-10k | MNIST-M | SVHN | USPS | SYNTH | DG Avg | MNIST-C |
|---|---|---|---|---|---|---|---|
| $RC_{mix1}$ | $98.62_{(0.06)}$ | $83.98_{(0.98)}$ | $53.26_{(2.59)}$ | $80.57_{(1.09)}$ | $59.25_{(1.38)}$ | $69.26_{(1.35)}$ | $88.59_{(0.38)}$ |
| $RC_{mix1-3}$ | $98.76_{(0.02)}$ | $84.66_{(1.67)}$ | $55.89_{(0.83)}$ | $80.95_{(1.15)}$ | $60.07_{(1.05)}$ | $70.39_{(0.58)}$ | $89.80_{(0.94)}$ |
| $RC_{mix1-5}$ | $98.76_{(0.06)}$ | $84.32_{(0.43)}$ | $\mathbf{56.50_{(2.68)}}$ | $81.85_{(1.05)}$ | $60.76_{(1.02)}$ | $70.86_{(0.86)}$ | $90.06_{(0.80)}$ |
| $RC_{mix1-7}$ | $98.82_{(0.06)}$ | $84.91_{(0.68)}$ | $55.61_{(2.63)}$ | $\mathbf{82.09_{(1.00)}}$ | $\mathbf{62.15_{(1.30)}}$ | $\mathbf{71.19_{(1.21)}}$ | $90.30_{(0.44)}$ |
| $RC_{mix1-9}$ | $98.81_{(0.12)}$ | $\mathbf{85.13_{(0.72)}}$ | $54.18_{(3.36)}$ | $82.07_{(1.28)}$ | $61.85_{(1.41)}$ | $70.81_{(1.24)}$ | $\mathbf{90.83_{(0.52)}}$ |
| $RC_{img1,\,p=0.5}$ | $98.66_{(0.05)}$ | $85.12_{(0.96)}$ | $55.59_{(0.29)}$ | $80.65_{(0.71)}$ | $60.85_{(0.48)}$ | $70.55_{(0.15)}$ | $89.00_{(0.45)}$ |
| $RC_{img1-3,\,p=0.5}$ | $98.79_{(0.07)}$ | $85.36_{(1.04)}$ | $\mathbf{55.60_{(1.09)}}$ | $80.99_{(0.99)}$ | $61.26_{(0.80)}$ | $70.80_{(0.86)}$ | $89.84_{(0.70)}$ |
| $RC_{img1-5,\,p=0.5}$ | $98.83_{(0.07)}$ | $\mathbf{86.33_{(0.47)}}$ | $54.99_{(2.48)}$ | $80.82_{(1.83)}$ | $62.61_{(0.75)}$ | $71.19_{(1.25)}$ | $90.70_{(0.43)}$ |
| $RC_{img1-7,\,p=0.5}$ | $98.83_{(0.07)}$ | $86.08_{(0.27)}$ | $54.93_{(1.27)}$ | $\mathbf{81.58_{(0.74)}}$ | $\mathbf{62.78_{(0.86)}}$ | $\mathbf{71.34_{(0.61)}}$ | $\mathbf{91.18_{(0.38)}}$ |
| $RC_{img1-9,\,p=0.5}$ | $98.80_{(0.12)}$ | $85.63_{(0.70)}$ | $52.82_{(2.01)}$ | $81.48_{(1.22)}$ | $62.55_{(0.74)}$ | $70.62_{(0.73)}$ | $90.79_{(0.48)}$ |

Table 10: Ablation study of consistency loss weight $\lambda$ on digits recognition benchmarks for $RC_{mix1-7}$ and $RC_{img1-7,p=0.5}$. DG-Avg is the average performance on MNIST-M, SVHN, SYNTH and USPS. Best results for each variant are **bold**.

| | $\lambda$ | MNIST-10k | MNIST-M | SVHN | USPS | SYNTH | DG Avg | MNIST-C |
|---|---|---|---|---|---|---|---|---|
| | 20 | $98.90_{(0.05)}$ | $87.18_{(0.81)}$ | $\mathbf{57.68_{(1.64)}}$ | $\mathbf{83.55_{(0.83)}}$ | $63.08_{(0.50)}$ | $72.87_{(0.47)}$ | $91.14_{(0.53)}$ |
| | 10 | $98.85_{(0.04)}$ | $\mathbf{87.76_{(0.83)}}$ | $57.52_{(2.09)}$ | $83.36_{(0.96)}$ | $62.88_{(0.78)}$ | $\mathbf{72.88_{(0.58)}}$ | $\mathbf{91.62_{(0.77)}}$ |
| $RC_{mix1-7}$ | 5 | $98.94_{(0.09)}$ | $87.53_{(0.51)}$ | $55.70_{(2.22)}$ | $83.12_{(1.08)}$ | $62.37_{(0.98)}$ | $72.18_{(1.04)}$ | $91.46_{(0.50)}$ |
| | 1 | $98.95_{(0.05)}$ | $86.77_{(0.79)}$ | $56.00_{(2.39)}$ | $83.13_{(0.71)}$ | $\mathbf{63.18_{(0.97)}}$ | $72.27_{(0.82)}$ | $91.15_{(0.42)}$ |
| | 0.1 | $98.84_{(0.07)}$ | $85.41_{(1.02)}$ | $56.51_{(1.58)}$ | $81.84_{(1.14)}$ | $61.86_{(1.44)}$ | $71.41_{(0.98)}$ | $90.72_{(0.60)}$ |
| | 0 | $98.82_{(0.06)}$ | $84.91_{(0.68)}$ | $55.61_{(2.63)}$ | $82.09_{(1.00)}$ | $62.15_{(1.30)}$ | $71.19_{(1.21)}$ | $90.30_{(0.44)}$ |
| | 20 | $98.79_{(0.04)}$ | $87.53_{(0.79)}$ | $53.92_{(1.59)}$ | $81.83_{(0.70)}$ | $62.16_{(0.37)}$ | $71.36_{(0.49)}$ | $\mathbf{91.20_{(0.53)}}$ |
| | 10 | $98.86_{(0.05)}$ | $87.67_{(0.37)}$ | $54.95_{(1.90)}$ | $82.08_{(1.46)}$ | $63.37_{(1.58)}$ | $72.02_{(1.15)}$ | $90.94_{(0.51)}$ |
| $RC_{img1-7,p=0.5}$ | 5 | $98.90_{(0.04)}$ | $\mathbf{87.77_{(0.72)}}$ | $\mathbf{55.00_{(1.40)}}$ | $\mathbf{82.10_{(0.55)}}$ | $\mathbf{63.58_{(1.33)}}$ | $\mathbf{72.11_{(0.62)}}$ | $90.83_{(0.71)}$ |
| | 1 | $98.86_{(0.04)}$ | $86.74_{(0.32)}$ | $53.26_{(2.99)}$ | $81.51_{(0.48)}$ | $62.00_{(1.15)}$ | $70.88_{(0.93)}$ | $91.11_{(0.62)}$ |
| | 0.1 | $98.85_{(0.14)}$ | $86.85_{(0.31)}$ | $53.55_{(3.63)}$ | $81.23_{(1.02)}$ | $62.77_{(0.80)}$ | $71.10_{(1.31)}$ | $91.13_{(0.69)}$ |
| | 0 | $98.83_{(0.07)}$ | $86.08_{(0.27)}$ | $54.93_{(1.27)}$ | $81.58_{(0.74)}$ | $62.78_{(0.86)}$ | $71.34_{(0.61)}$ | $91.18_{(0.38)}$ |

## F    MORE EXAMPLES OF RANDCONV DATA AUGMENTATION

We provide additional examples of `RandConv` outputs for different convolution filter sizes in Fig. 6 and for its mixing variants at scale $k = 7$ with different mixing coefficients in Fig. 5. We observe that `RandConv` with different filter sizes retains shapes at different scales. The mixing strategy can continuously interpolate between the training domain and a randomly sampled domain.

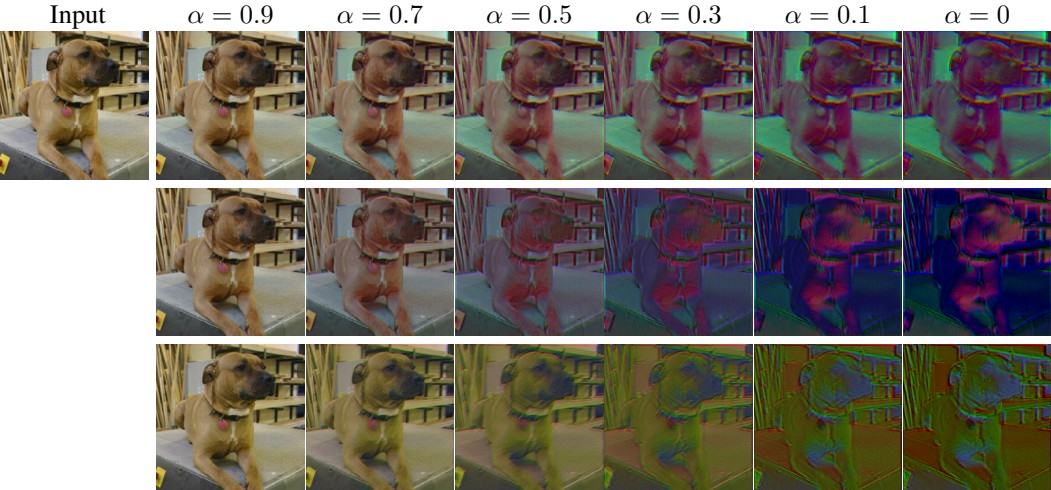

Figure 5: Examples of the `RandConv` mixing variant $RC_{mix7}$ on images of size $224^2$ with different mixing coefficients $\alpha$. When $\alpha = 1$, the output is just the original image input;when $\alpha = 0$, we use the output of the random convolution layer as the augmented image.

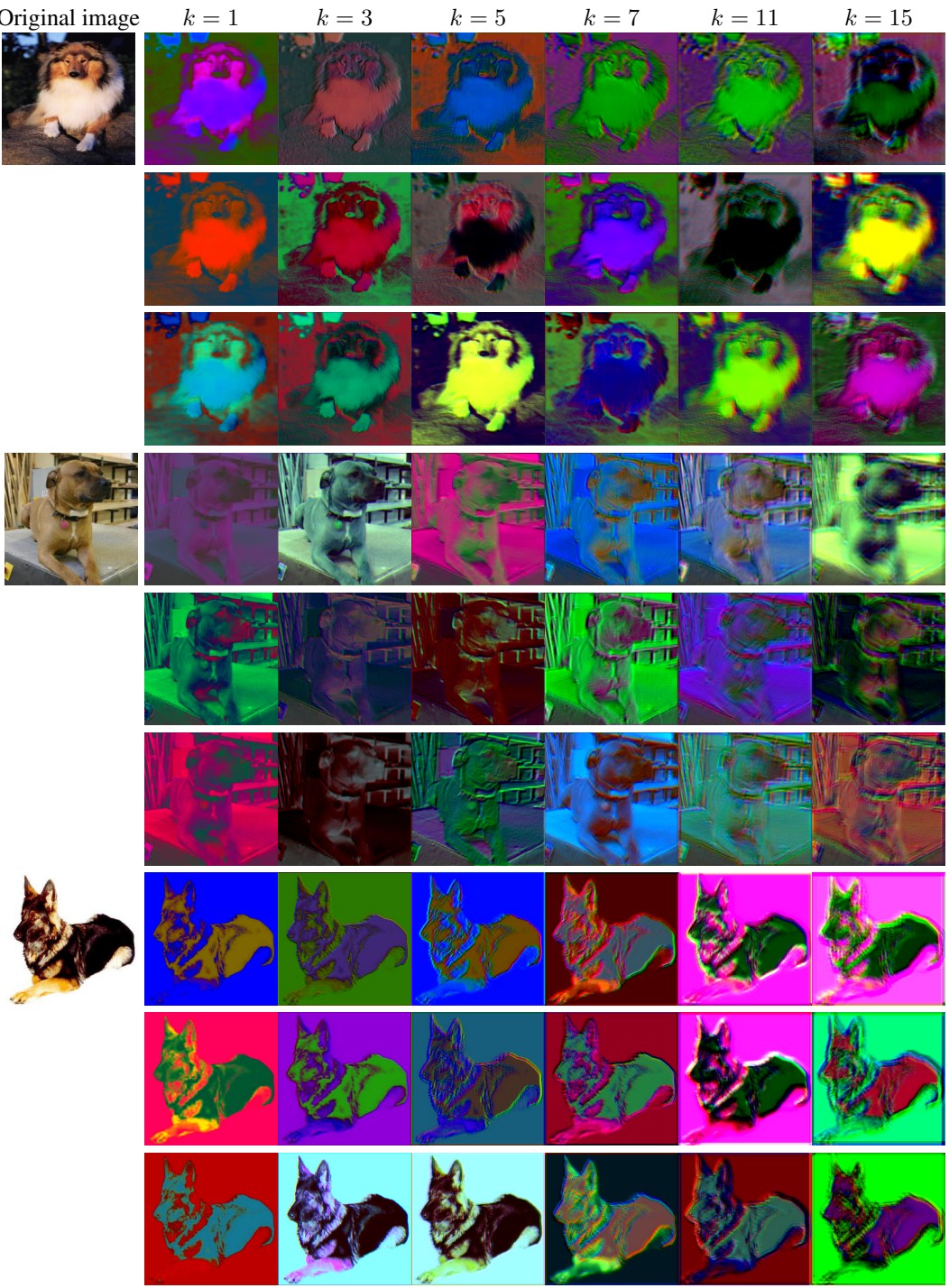

Figure 6: RandConv data augmentation examples on images of size $224^2$. First column is the input image; following columns are convolution results using random filters of different sizes $k$. We can see that the smaller filter sizes help maintain the finer shapes.

