# OpenReview forum: "Robust and Generalizable Visual Representation Learning via Random Convolutions"
_ICLR.cc/2021/Conference — ICLR 2021 Poster_

### Official Review · AnonReviewer4 · 2020-10-27
**Why this particular approach and what about colour?**

**Rating:** 6
**Confidence:** 4

**Review:**

Update
-------

I've updated by score in light of the discussion; as I said in the comments, from a purely experimental point of view there are good results, however the presentation of the paper confounds too many aspects. If the authors can address the terminology issues then it would make the work stronger.


Summary
-------

This paper proposes the use of a random convolution layer at the beginning of a network during training. The weights of this layer are not learned, and are changed randomly each time an input is passed through. In the experiments presented the layer has 3 input and output channels, so that the effect of the convolution is to jumble-up colour and modify local texture (at a scale determined by the size of the convolution kernel, which is itself uniformly sampled from a set of possible scales). During training the objective is modified with an additional loss term that intends to ensure that the distribution of network outputs from three independent forward passes of the network with the random convolution are consistent with each other. The normal task loss is just computed on the basis of the output from one of the forward passes.  The authors posit that this approach will lead to increased robustness and generalisation of the network as if forces a bias towards shape and away from texture (and colour), and demonstrate this with a wide range of experimental results.


Positives
---------

- A wide range of metrics and datasets has been used to validate the performance of the proposed approach, and the evaluation appears thorough with respect to comparing against state-of-the-art approaches.
- Aside from a slight lack of detail on the testing regime (see questions below), which is easily corrected, the paper itself reads very well.
- The results do show that the proposed approach can have significant performance gains in a number of tasks.


Concerns
--------

- The proposed method confounds texture and colour; these are different things and should be treated as such. One of the likely reasons for the reduction in performance on photos is for classes where colour is important to choosing the correct class. This probably also explains why mixing might be important/useful in some cases. I'm confident that if the authors had addressed this they would see further improvements in their results.
- Whilst I agree that the random convolutions would obviously bias a network to learn a shape bias due to the destruction of local texture and colour information, using random convolutions to do this seems incredibly arbitrary given the myriad of possible ways of achieving this. For example, a trivial way to achieve the same thing would be to use a bandpass filter such as a Laplacian of Gaussian (coupled with random mixing of the colour channels if you really wanted to remove the colour information too). Such an approach could also trivially be made multi-scale if desired, and would be justifiable (e.g. Marr from a neuroscience perspective; Lindeberg from scale-space theory).
- No adequate baseline has been provided; following on from the above point, I would have at least expected experimental comparison against a _simple_ baseline to demonstrate the advantages of the proposed technique (e.g. using a fixed bandpass filter instead of a random convolution perhaps).


Rationale for score
-------------------

As it stands, although this paper clearly provides a method that gives good results, I find it difficult to give a high score because of the arbitrariness of the proposed method and the lack of justification for the approach (other than "it seems to work"). I am also strongly concerned by the way the paper confuses texture and colour.

Questions during rebuttal period
--------------------------------

- Just to confirm: what is the testing set-up used? Do you input the raw image data into the network without passing through the RandConv?
- Have you measured the shape bias (Geirhos'19) of networks trained using RandConv? Do the networks show bias to shape over texture using this metric?

---

> ### Author Response · Authors · 2020-11-18
> **Color can hurt domain generalization as much as texture does; RandConv was justified in appendix; the suggested baseline is tested.**
>
> Thank you very much for your comments and suggestions. We try to address your questions and concerns as follows:
>
> Questions
> 1. *What is the testing set-up used?* \
> 	We did not apply RandConv to inputs during testing, we only use RandConv during training. Test-time augmentation would be an interesting study for future work.
>
> 2. *Have you measured the shape bias (Geirhos'19) of networks trained using RandConv? Do the networks show bias to shape over texture using this metric?* \
> 	Thanks for this suggestion, we now tested the ImageNet trained AlexNet using the shape bias metric in Geirhos'19 as an example to illustrate the shape-bias of RandConv. The shape-biases of AlexNet trained with RandConv are improved over the baseline, shown in the following table. We will add this to the final manuscript.
>  |  | Baseline | RC-img,p=0.5 | RC-mix | RC-img,p=0 |
>  |-|-|-|-|-|
>  | Shape Bias (%) | 25.36 | 48.24 | 54.85 | 61.66 |
>
> Concerns
> 1. *Texture v.s. Color*
> - Thanks for pointing out the terminological confusion. Since our work is inspired by the "texture bias" discovery in Geirhos'19, we follow their terminology regarding "shape" and "texture" where the texture changes include color changes. We will clarify this in the revised version.
> - Geirhos'19 and our work choose to disrupt both texture and color since neither is an appropriate prior cue for determining the semantic classes of images. Biases towards either color or texture can hurt generalizability in new domains. As discussed in Sec.5 of the paper, color and texture can both be helpful in some cases, e.g. for fine-grained visual recognition. Then, it becomes a trade-off between texture/color bias and shape bias for out-of-domain generalizability v.s. in domain performance.
> - We point to sec 4.4, where we explore the reason for and solution to the Photo performance drop, shown in Table 2. In short, the texture bias in the pretraining model leads to the performance drop when fine-tuning with the shape bias. Using a shape-biased pretrained model can further improve the generalizability and maintain the performance in domains where color bias can be useful, e.g. Photo, shown in Table 4.
>
> 2.  *RandConv sounds arbitrary. Why not use a bandpass filter for the destruction of local textures.*
> - We agree that a proposed method should have a clear motivation and be well justified. **Detailed justifications on why RandConv preserve shapes are provided in Appendix A and B.** Using the Johnson–Lindenstrauss lemma, we show that shapes are maintained by RandConv thanks to the relative distance preserving property of random linear projections. We also note that, strictly, a convolution layer is different from the convolution operation in image filtering. Image filtering applies the same 2D filter on three color channels separately while our convolution layer applies three different 3D filters and each takes all color channels as input and generates one channel of the output.
> - A random convolution is expected to be better than applying a fixed filter, e.g. a bandpass filter, since it can create outputs with similar shapes but more diverse texture/color styles; see Figs. 1 and 6 in the appendix. In contrast, a fixed filter produces outputs with a specific texture style which contradicts the goal of data augmentation of providing diversity and therefore could lead to overfitting.
> - We appreciate the suggestion of using a fixed Laplacian of Gaussian filter (LoG) as a baseline. We now conducted such an experiment using LoG filtering with size 3 and sigma=1. On the digits data, a fixed LoG improves the DG-Avg to 63.58% but still shows worse performance than RandConv's 71.34% without mixing and the consistency regularization. The LoG approach does not benefit MNIST-C nor PACS. We will provide more details in the revised version.

---

> > ### Comment · AnonReviewer4 · 2020-11-24
> > **Follow-up comments**
> >
> > Dear Authors,
> >
> > Thank you for your responses and additional experiments. Like Reviewer 1, I'm still a bit on the fence with this paper because I believe it still confounds two very distinct aspects (I'm in total agreement that to some extent Geirhos' work also did the same, but that isn't an excuse to properly address this issue). I would also strongly disagree with your statement that neither colour or texture is an appropriate cue for distinguishing semantic classes. How for example, could one tell the difference between a horse and zebra without texture? Regarding colour, do you really think that you can get really good performance on datasets like CLEVER which has tasks requiring reasoning over coloured objects, or CUB in which the fine-grained classification task requires strong colour cues (at least as far as humans are concerned). Of course there are also many counter-examples too this...
> >
> > In terms of the justification, I feel that you are missing the point to some extent - random convolutions with values of mixed signs are going to in expectation behave like very unstructured bandpass filters - this fits in with the proof in the appendix. This doesn't however address why this approach is better than using more structured filters (with some variability built-in, perhaps in terms of bandwidth, or just in terms of noise [like in the Dapello  paper from NeurIPS 2020]). I'm also totally aware that in image processing the convolutions are typically applied channel-wise, whereas here each random convolution spans the channels, however one can view the behaviour of _each_ of your random convolutions as applying a different bandpass filter to each channel and to sum the results into a single output. I think what both Reviewer 1 and I really want to see is some strong justification as to why this approach either reasonable (perhaps in terms of mimicking something in biology or otherwise) or is better than a more structured approach.
> >
> > I think perhaps the real problem I have is that you're confounding too many things - a randomised augmentation, an approach to improve shape bias and an approach that destroys useful colour information and putting them all under the same umbrella and terminology. Like R1, I'm not against this paper getting accepted on the basis of the empirical results that are already present, however I do believe that it would be a much better paper if it more thoroughly disentangled the underlying parts and used better terminology to describe what is actually going on.

---

> > > ### Author Response · Authors · 2020-11-24
> > > **Re: Follow-up comments**
> > >
> > > - We are happy to change terminology in a final version to make it more explicit what our augmentation strategy is based on and that it includes color changes. Our initial intent was to stay consistent with the terminology introduced in Geirhos et al., but we agree that being more upfront about what is and what isn’t included in our random augmentations would be better. If you have another term in mind, please let us know and we will be happy to use it instead. For example, we would be happy to change our terminology to **combined texture and color randomization** (which would make the color randomization more explicit) or **style randomization** (though style might be a little too generic and overloaded as well). We are, of course, also open to other suggestions.
> > >
> > > - We did not mean to imply that texture or color is in general not useful features. In fact, as you point out, they might be very relevant. But the utility of texture and color will depend on what domains we want to generalize between. Note also that our notion of randomization is randomization at a particular scale. Hence, for the zebra example, given a filter that is not too large, one would retain the stripes but would remove the fine-scaled structure. In fact, the stripes would be basic shapes in our formulation. Further, by randomizing over color we reduce sensitivity to color cues and might therefore be able to make a zebra classifier less sensitive to lighting (think of a zebra at sunset versus one in broad daylight) and styles (e.g., photograph versus sketch). Our intuition is that texture/color may result in a model overlooking shape cues, even if image semantics considered natural for humans should focus on shape. E.g. a texture-biased model may confuse a zebra-textured backpack with a zebra. There are, of course, cases where both color or texture (at a particular scale) might matter. For example, if one would like to distinguish white from black horses, color information will be crucial and our current RandConv would be detrimental to performance. In summary, there will be a trade-off between generalization and performance that will also depend on what domains one would like to generalize across. We discussed such limitations in the manuscript but will add a more in-depth discussion to further clarify these points.
> > >
> > >
> > > - Disentangling color and texture as well as other randomization approaches would indeed be interesting. We cannot speculate at this point on how such approaches would in general compare to RandConv (other than that RandConv worked better than the Laplacian of Gaussian approach we added to the manuscript in response to the initial review). But we would be very interested in exploring this in future work and are happy to point out these opportunities in a final version of the manuscript. For us, an attractive feature of RandConv is that it is simple yet effective. A more structured approach could have the benefit of being more adapted to a particular setting, but it would likely come at the cost of being less generically applicable.

---

### Official Review · AnonReviewer1 · 2020-10-28
**Interesting use of Random convolutions for Data-Augmentation**

**Rating:** 6
**Confidence:** 4

**Review:**

This paper proposes a simple way to increase the robustness of the learned representations in a network perform a series of object recognition tasks by adding a random convolution layer as a pre-processing stage, thus “filtering the image” and preserving the global shape but altering the local `texture’ of the newly transformed image. Here, the hope is that  -- analogous to Geirhos et al. 2019 that induces a shape bias by transforming the image distribution into a new one with altered *global* textures that induce a shape bias and increases general robustness to o.o.d distortions --  the authors here go about doing something similar at the local level given the small size of the receptive field of the filter, thus preserving the shape and slightly altering “the texture”.

Pros:
* While the innovation is simple and efficient, this data-augmentation scheme works, and I can see how other future works may use this as well as a data-augmentation technique for object recognition. I am not sure however if no one else has explored the effects of random convolutions for robustness. It sounds too good to be true, but then again -- there is always beauty in simplicity and it is possible that the authors have hit the nail on the head on finding a somewhat ‘contrived’ filtering process as a bonus rather than a limitation. Simple, yet counter-intuitive findings like these are relevant for ICLR.
* Authors provide lots of experiments that to some degree prove the success of their augmentation strategy (although see Cons).

Cons:
* Biological Inspiration: What is the biological mechanism linked to the success of using random convolutions. One could argue that this point is ‘irrelevant’ to the authors and the readers, but as there is a plethora of different data-augmentation techniques to choose from, why should computer vision and machine learning practitioners choose this one? (See Missing Reference for a suggestion)
* Insufficient/Incomplete Baseline: The model is inspired loosely by Geirhos et al. 2019; but how does the model compete with Geirhos’ et al.’s Stylized ImageNet? I would have wanted to see a baseline between the authors proposed model and other texture-based augmentation strategies. This would elucidate the Global vs Local advantages of “texture”/style transfer on learned representations. I think this is where authors could capitalize more on.
* The word `texture’ in the paper is a mis-nomer. Here what is really done is 1st order filtering via a convolution operation with a filter that does not happen to have a Gabor-like shape. “Texture” in other contexts going back to vision science and even computer vision and image processing (style transfer included), is usually computed by a set of cross-correlations between *outputs* of a filtered image (analogous to the Gramian Matrix of Gatys et al. 2015), or the principled Portilla-Simoncelli texture model from 1999.

Missing references:
* Excessive Invariance increases adversarial vulnerability by Jacobsen et al. ICLR 2019. The augmentation procedure proposed by the authors shows robustness to common distortions, but how about adversarial robustness? Is this relevant? Was this tried? I’d love to hear more about the authors thoughts on this to potentially raise my score.
* Emergent Properties of Foveated Perceptual Systems (link: https://openreview.net/forum?id=2_Z6MECjPEa): An interesting concurrent submission to this year's ICLR has shown that the biological mechanism of visual crowding (that resembles texture computation for humans in the visual periphery) is linked to some of the operations introduced in the paper by the authors. It would be great if the authors potentially cite similar (and/or the before-mentioned) works to provide a link to a biological mechanism that may support why their data-augmentation procedure works and/or should be used; otherwise it seems contrived and could be seen as “yet another data-augmentation procedure that increases robustness but we don’t know why”.
* Implementing a Primary Visual Cortex in the retina increases adversarial robustness by Dapello, Marques et al. 2020 (NeurIPS). This recently published paper in a way shows almost the opposite of what the authors are proposing here. Rather than using random convolutions, they actually mimic the gamut of spatial frequency tuning properties of Gabor filters in the first stages of convolution as done in human/monkey V1. The authors should discuss how their results fit with Dapello, Marques et al. 2020 and how they can reconcile their somewhat opposing views.

Final Assessment:
I am on the fence of having this paper accepted at ICLR given the limitations expressed above, but I do like it’s simplicity that should not take away it’s merit -- thus my slight lean towards acceptance. I am willing to raise my score however if authors address some of the cons/limitations, and am also curious to see the opinion from other reviewers, it is possible that I may have missed a key reference regarding data-augmentation that may weaken my assessment.

---

> ### Author Response · Authors · 2020-11-18
> **RandConv was justified by relative distance preserving of linear random projections proved in Appendix; Stylized ImageNet is tested as suggested (Part1)**
>
> Thank you for recognizing the simplicity and effectiveness of our method, as well as for the thoughtful comments and suggestions. Please find detailed answers to the raised concerns below.
>
> 1. *What is the biological mechanism linked to the success of using random convolutions? RandConv seems contrived and could be seen as “yet another data-augmentation procedure that increases robustness but we don’t know why”.*
> - We agree with you that connecting our design with the biological mechanism could be interesting, and will add a note to our revised version about doing this in future work. However, we do have clear motivation and justification on why we want to use RandConv and why it works. It is as follows:
> - Inspired by Geirhos et al. 2019 and by our observation that consistent features across visual domains are mostly shapes, we are motivated to introduce a shape bias to a model for better generalizability and robustness. Data augmentation has been a common way to induce ML model invariance to some features, e.g. texture and color in our case, by diversifying these features without changing labels.
> - Certain filter classes, e.g., Gaussian filters, Sobel filters, etc.,  produce images with distinct filter-specific fixed styles. However, we further showed that random convolution layers can preserve shapes and randomly disrupt texture. **We justified this in Theorem 1 in the Appendix.** The formulation and proof of Theorem1 are inspired by the Johnson–Lindenstrauss lemma on the distance-preserving property of random linear projections, which has been widely used in machine learning applications.
>
> 2. *How does the model compete with Geirhos et al.’s 19 Stylized ImageNet (SIN)?*
> - We now follow your suggestion and test the SIN-trained AlexNet provided by Geirhos et al. for A) the ImageNet-sketch benchmark and B) as a pretrained model for finetuning on the PACS benchmark with the DeepAll baseline. Our results show that these approaches achieve comparable performance to RandConv, however, at the cost of a much more complex model (see details below). Specifically, for A) SIN-AlexNet obtains 17.62% Top1 accuracy, which is lower than RC_img(18.09%) and higher than RC_mix(16.91%). For B) SIN-AlexNet improves the DeepAll baseline to 70.58(0.21%) while RC_img and RC_mix achieve 70.61%and 70.25% respectively. We will provide more details in the revised paper.
> Intuitively, the advantage of image stylization is that it can modify image texture in a hierarchical and realistic way, but as shown in the above experiments SIN still performs on par with RandConv. Note that image stylization techniques require additional data and heavy precomputation. Further, the images for the style source also need to be chosen. In contrast, RandConv is much easier to use: it can be applied to any dataset via a simple convolution layer.
>
> 3. *The word 'texture' in the paper is a misnomer:* \
> Thank you for bringing up this concern. RandConv with a three-channel output can be seen as generating new images with the same shape but different “style”. We loosely refer to this property as texture, but do not intend to use RandConv as a model/filter to represent or parameterize texture in the ways discussed in the review. We will clarify this in the revised version.
>
> 4. *Is adversarial robustness relevant and has it been tested?* \
> Zhang & Zhu 2019 [1] find that adversarially-trained models are more shape biased and Shi et al. 2020 [2] show that their method for increasing shape bias also helps adversarial robustness, especially when combined with adversarial training. Hence, we believe that adversarial robustness is definitely a relevant direction and RandConv could potentially be helpful. However, our work is focused on domain generalization and robustness to common corruptions. Therefore, we leave adversarial robustness for future work. We will add a discussion on the potential for improving adversarial robustness to our revised draft.
>
> [1] [Interpreting Adversarially Trained Convolutional Neural Networks ICML2019](https://arxiv.org/pdf/1905.09797.pdf)\
> [2] [Informative Dropout for Robust Representation Learning: A Shape-bias Perspective  ICML2019](https://arxiv.org/pdf/2008.04254.pdf)

---

> > ### Author Response · Authors · 2020-11-18
> > **RandConv was justified by relative distance preserving of linear random projections proved in Appendix; Stylized ImageNet is tested as suggested (Part2)**
> >
> > 5. *Missing references* \
> > 	Thank you for providing interesting relevant work. We have addressed the questions related to the first work on adversarial robustness in 4 and the second work on biological motivations in 1. We think the third work by Dapello, Marques et al. 2020 shares similar intuitions with our RandConv. Although they use Gabor filters and we use random convolution kernels, their filters are still randomly sampled from a Gabor filter bank and there is a stochastic layer in their design which may motivate the importance of randomness. However, the connection with our methods is not straightforward due to a key difference: our RandConv works as data augmentation (i.e. generating new training samples) and is not used during testing; the design in Dapello, Marques et al. 2020 is a non-learnable block that is part of the whole neural network and remains present during testing. Moreover, image filtering by applying separate convolutions on the different color channels is different from a convolution layer, where one considers sums of convolutional responses over all input channels for each output channel. We will add discussions on this related work to the revised version of our manuscript.

---

> > ### Comment · AnonReviewer1 · 2020-11-24
> > **Follow up comments [Additional concern: Texture vs Color]**
> >
> > Dear Authors,
> >
> > Thank you for responding to all my previous concerns.
> >
> > While the method empirically works, I am still on the fence with this paper given the terminology (what is referred here as texture, again is in fact a misnomer and is something that Reviewer 4 has pointed out by finding color as a confounding variable). Unless I missed something, I don't think I've seen a new random colorization baseline/control that will give support to use the word 'texture' in the paper. However, I may have missed this and if so, please let me know where I should look.
> >
> > As a side-note, perhaps I am a bit nosy with the terminology (and so is reviewer 4), but I think it is important as it could lead to future mis-conceptions of why this data-augmentation procedure succeeds. The paper could get accepted as is given the fact that it empirically works and the added baselines strengthen several results  (I am just as enthusiastic as R2 and R3 on this), but correcting the terminology -- and not making the far-stretched claim that first order random filtering is equivalent to texturing an image -- would still be important.

---

> > > ### Author Response · Authors · 2020-11-24
> > > **Re: follow-up comments**
> > >
> > > - For consistency, we started with the terminology by Geihros et. al., where texture includes color (as also acknowledged by R4) and had opted for the clarification that RandConv changes both color and texture in the initial revised manuscript. However, we realize that texture is usually thought of independently of color. As our goal is a clear and descriptive terminology we are happy to change our used terminology. Please let us know if you have another term in mind and we will be happy to use it. For example, we would be happy to change the terminology to **combined texture and color randomization** (which would make the color randomization more explicit) or to **style randomization** (though style might be a little too generic and overloaded as well). We are, of course, also open to other suggestions.
> > >
> > > - As far as a new random colorization baseline/control is concerned, this is included in our paper under the name **“ColorJitter”**. The ColorJittering results are given in Tables 1 and 2.

---

### Official Review · AnonReviewer2 · 2020-10-28
**A novel and effective data augmentation method**

**Rating:** 7
**Confidence:** 3

**Review:**

Summary: This paper describes a method 'RandConv' which uses random convolutions to generate images with random textures but maintain global shape to improve training and generalization for classification tasks. The authors compare both random convolutions as images and mixing them with original training images.

Strengths:
The authors demonstrate success with their method across various datasets as well as for pre-training experiments and evaluating on fine-tuning datasets.

The authors describe why their method should work as well as why their random convolutions preserve global shape. Far too often in ML papers we stop at 'this worked' and do not take the time to understand the why or how.

The authors compare their methods to other data augmentation methods and demonstrate improvements.

The t-SNE visualization of the difference in training is great evidence for this work and very clear from a reader standpoint.

Weaknesses:
The authors neglect to compare across a significant amount of data augmentation methods, making it less clear how their method would compare.

Their method has worse performance in their PACS experiments for the Photo category and results vary significantly based on some algorithm choices in using their RandConv methods. For this to be more useful, a clearer understanding of the use cases is needed.

Minor comment: typo on page 5 'w train a simple CNN' under 4.1 paragraph 1

The authors present a novel and interesting approach for data augmentation that can be directly coded into CNNs and demonstrate improvement for some tasks. I recommend accepting this work, although more work needs to be done to understand when and how to use their methods.

---

> ### Author Response · Authors · 2020-11-18
> **Baseline with extra data augmentations added**
>
> Thank you very much for appreciating our efforts in showing how and why RandConv works. Your questions and concerns are addressed below:
>
> 1. *Comparison with other data augmentation methods.*
> - Although there are many data augmentation methods, it is more sensible to compare with those that are helpful for generalization or may increase shape bias. Therefore, we compared with colorjitter and grayscale which can bias the model to shape cues; in the digits datasets, we also compare to more advanced adversarial data augmentation methods GUD and M-ADA since they were proposed for generalization and robustness. RandConv outperforms these methods on domain generalization tasks.
> - In addition, we now also test a data augmentation pipeline that was used in a recently proposed large-scale study on domain generalization algorithms and datasets [1]. This data augmentation pipeline can improve the baseline performance on both digits datasets and PACS datasets. However, results are still inferior to RandConv. When combined with the data augmentation pipeline, RandConv results in improvements for the digits dataset but not for PACS. See tables in the [comment](https://openreview.net/forum?id=BVSM0x3EDK6&noteId=IzZersT2KAU) to reviewer#3 and details in the revised manuscript.
>
> 2. *Performance drops when generalizing to the Photo domain and is sensitive to algorithm choice.* \
> We point to sec 4.4, where we explore the reason for and solution to the Photo performance drop, shown in Table 2. In short, the texture bias in the pretraining model leads to the performance drop when fine-tuning with the shape bias. Using a shape-biased pretrained model can further improve the generalizability and maintain the performance in domains where color bias can be useful, e.g. Photo, shown in Table 4.  In fact, our algorithm is robust to the choice of the hyperparameter and consistency regularization, shown by the ablation study in Figure 2.
>
> 3. *A clearer understanding of the use cases is needed* \
> It would help computer vision tasks where a shape biased model is helpful e.g. object detection. As shown by Sec 4.4, our method can provide a shape-biased pretrained model to improve performance on downstream tasks when generalizing to unseen domains. We will add some discussion of use-cases to our revised version.
>
> 4. *Typo on page 5 'w train a simple CNN' under 4.1 paragraph 1* \
> Thanks for pointing this out, we have fixed this typo in the revised version.
>
>
> [1] [In Search of Lost Domain Generalization](https://arxiv.org/abs/2007.01434)

---

### Official Review · AnonReviewer3 · 2020-10-30
**Interesting Method, Several Issues in Experiments**

**Rating:** 6
**Confidence:** 3

**Review:**

Summary: This paper uses random convolution augmentation to learn representation that is robust for transferring across domains.

Quality: The paper is well-written, easy to following. The method is presented clearly, and the enough experimental evaluation was conducted. Overall, the quality is good.

Clarity: The method is straightforward, and the authors describe it clearly. I think the idea is reasonable, but there is something to clarify in experiments:
1. Compared with [1], all the methods have a drop in Table 2. Could you explain why? In fact, Table 2 is really confusing, the bolded numbers are not the best numbers.
2. I notice that in [1], the performance of PAR can be further improved with strong data augmentation. In Table 2, the method is compared with Grey Scale/ Color Jitter. I wonder if this is fair comparison, since RandConv composes various random convolution schemes and mix. I wonder how the performance will change when the baseline uses stronger augmentation, for example, combining multiple data augmentation like self-supervised learning [2].
3. Just wondering: will your method result in a network with higher robustness against adversarial attacks?

Originality: As far as I know, the method is new.

Significance: I think this paper proposes a simple yet effective method. The use of random convolutions is interesting. I assume this work can be a good contribution to the community if they can refine their experimental evaluation to better justify the effectiveness of their method.

Reference:
[1] Learning Robust Global Representations by Penalizing Local Predictive Power https://arxiv.org/pdf/1905.13549.pdf
[2] A Simple Framework for Contrastive Learning of Visual Representations https://arxiv.org/abs/2002.05709

---

> ### Author Response · Authors · 2020-11-18
> **Methods with different DeepAll baselines are not directly comparable; experiments with extra data augmentations have been added.**
>
> We appreciate the positive feedback on the quality of our work. Our response to the individual experimental concerns are given below:
> 1. *In Table 2, why is PAR’s accuracy higher than other methods? How to interpret the bold numbers*
> - Unfortunately, previous approaches used different DeepAll baselines which make the final accuracy not directly comparable. For example, PAR [1] used a pretrained AlexNet implemented differently (with the local response normalization) that boosted the DeepAll baseline to 70.16%. JiGen [3] uses **different train/test splits** than the official ones and thus the test accuracy is not comparable. These discrepancies mean we cannot compare methods that use different DeepAll baselines. The bold numbers in Table 2 are therefore the best performance under our baseline. We will reorganize Table 2 to clarify this in the revised version.
> - We were not able to reproduce PAR’s reported performance both when using the provided PAR code or based on our reimplementation. The reason might be that the reported PAR performance is based on a single run. We, instead, run every experiment 5 times using the same set of random seeds.
> 2. *Will adding more data augmentations make RandConv even better like PAR showed?*
> - PAR [1] was tested on top of the DeepAll baseline of JiGen [3] which produces better performance. However, the baseline of JiGen not only includes extra grayscale augmentation but also uses different training schemes, most importantly **different train/test data splits.** Again, those numbers are not directly comparable due to different baseline implementations. Also, running PAR's code by 5 repeats, we can not reproduce the improvement of PAR on JiGen's baseline reported in [1].
> - We agree that it is meaningful to see how stronger augmentation changes the DeepAll baseline and RandConv. Based on your suggestion, we now test the augmentation pipeline in SimCLR [2] on our baseline but it reduced the average performance on PACS from 66.57 to 64.25. We suspect that this augmentation pipeline (which was designed for self-supervised learning) is too aggressive for our supervised learning setting. As the SimCLR augmentation pipeline did not yield improvements, we now also test a data augmentation pipeline that was used in a recently proposed large scale study on domain generalization algorithms and datasets [4]. Using this data augmentation pipeline improved the baseline performance on both digits datasets and PACS datasets while the results are still inferior to RandConv. When combined with the data augmentation pipeline, RandConv achieves improved performance on the digits dataset but not on PACS. See results in the tables below and we will provide more details in the revised version.
>
>  Results on digits datasets:
>
>  |  | mnist_m | svhn | usps | synth | avg | mnist-c |
>  |-|-|-|-|-|-|-|
>  | baseline | 58.87(3.73) | 33.41(5.28) | 79.27(2.70) | 42.43(5.46) | 53.50(4.23) | 88.20(2.10) |
>  | baseline + aug | 62.32(0.66) | 39.07(0.68) | 79.31(1.02) | 46.48(0.80) | 56.79(0.34) | 89.54(0.11) |
>  | RC-mix | 87.76(0.83)  | 57.52(2.09) | 83.36(0.96)  | 62.88(0.78) | 72.88(0.58) | 91.62(0.77) |
>  | RC-mix + aug | 87.89(0.29) | 62.07(0.62) | 84.39(1.02) | 63.90(0.63) | 74.56(0.46) | 91.40(0.93) |
>
>  Results on PACS:
>
>  |  | P | A | C | S | Avg |
>  |-|-|-|-|-|-|
>  | baseline | 86.77(0.42) | 60.11(1.33) | 64.12(0.32) | 55.28(4.71) | 66.57(1.36) |
>  | baseline + aug | 85.21(0.47) | 59.51(0.38) | 62.88(1.01) | 61.67(0.76) | 67.32(0.23) |
>  | RC-mix | 86.60(0.67) | 61.74(0.90) | 64.05(0.66) | 69.74(0.66) | 70.53(0.25) |
>  | RC-mix + aug | 86.23(0.74) | 61.91(0.76) | 62.69(0.76) | 67.74(1.21) | 69.64(0.49) |
>
> 3. *Does RandConv help adversarial robustness?* \
> Zhang & Zhu 2019 [5] find that adversarially trained models are more shape-biased and Shi et al. 2020 [6] show that their method for increasing shape bias also helps adversarial robustness, especially when combined with adversarial training. Based on these works, we believe RandConv has the potential to be helpful. However, as our work is focused on domain generalization and robustness to common corruptions, we will investigate robustness to adversarial perturbations in future work. We will add a discussion of improving adversarial robustness to our revised draft.
>
> References: \
> [1] [Learning Robust Global Representations by Penalizing Local Predictive Power NeurIPS 2019](https://arxiv.org/pdf/1905.13549.pdf)\
> [2] [A Simple Framework for Contrastive Learning of Visual Representations ICML2020](https://arxiv.org/abs/2002.05709)\
> [3] [Domain Generalization by Solving Jigsaw Puzzles CVPR2019](https://arxiv.org/pdf/1903.06864.pdf)\
> [4] [In Search of Lost Domain Generalization](https://arxiv.org/abs/2007.01434)\
> [5] [Interpreting Adversarially Trained Convolutional Neural Networks ICML2019](https://arxiv.org/pdf/1905.09797.pdf)\
> [6] [Informative Dropout for Robust Representation Learning: A Shape-bias Perspective  ICML2019](https://arxiv.org/pdf/2008.04254.pdf)

---

### Public Comment · ~Dinghuai_Zhang1 · 2020-11-11
**Related ICML work**

Hi authors, I really enjoy reading your submission. I'm writing the comment to introduce our *highly* related ICML work:

Shi B. et al, Informative Dropout for Robust Representation Learning: A Shape-bias Perspective, ICML2020. https://arxiv.org/abs/2008.04254

Hope you could add this reference into your submission :).

---

> ### Author Response · Authors · 2020-11-18
> **We will add your work into our references.**
>
> Thanks for pointing your interesting work to us! We will add the reference in the revised version.

---

### Author Response · Authors · 2020-11-21
**To all reviewers:**

Thank you all for providing thoughtful comments and constructive suggestions. We appreciate your positive feedback on the novelty, simplicity, and effectiveness of RandConv and regarding the quality of our manuscript. We have replied to each review individually to address concerns and questions. Based on all reviews, we have now revised our manuscript. Specifically, we have made the following changes:

1. We added clarifications on:
    - the results and comparison in Table 2 (R3):
    - the “texture” terminology. (R4 & R1)

2. We added more experiments:
    - we added  the data augmentation pipeline in [1] to our baseline and RandConv (R2 & R3)
    - we now compare with using fixed Laplacian of Gaussian filtering for data augmentation (R4)
    - we now compare to the  Stylized ImageNet trained model in Geirhos'19 (R1)
    - we now measure the shape bias metric in Geirhos'19 on our RandConv-trained ImageNet model (R4)

3. We added discussions on:
    - the use cases of RandConv (R2)
    - connections with adv robustness  (R1)
    - exploring connections with biological mechanisms as future work (R1)

Finally, we would like to highlight **the justification on how and why RandConv works**, provided in Appendix A and B, as well as section 4.4 that explores **how the shape-bias of a pre-trained model can affect the generalizability of models in downstream tasks**, which explains why RandConv results in a performance drop on the Photo domain when using the baseline ImageNet model.

[1] [In Search of Lost Domain Generalization](https://arxiv.org/abs/2007.01434)bs/2007.01434)

---

### Decision · Program_Chairs · 2021-01-07
**Final Decision**

**Decision:**

Accept (Poster)

**Comment:**

Reviewers concurred that this is an interesting paper with contributions worthy of publication. The authors also provided many details in the rebuttal which makes the paper even more strong.